# Discriminative Feature Attributions: Bridging Post Hoc Explainability and Inherent Interpretability

**Usha Bhalla***
Harvard University
usha_bhalla@g.harvard.edu

**Suraj Srinivas***
Harvard University
ssrinivas@seas.harvard.edu

**Himabindu Lakkaraju**
Harvard University
hlakkaraju@hbs.edu

## Abstract

With the increased deployment of machine learning models in various real-world applications, researchers and practitioners alike have emphasized the need for explanations of model behaviour. To this end, two broad strategies have been outlined in prior literature to explain models. Post hoc explanation methods explain the behaviour of complex black-box models by identifying features critical to model predictions; however, prior work has shown that these explanations may not be faithful, in that they incorrectly attribute high importance to features that are unimportant or non-discriminative for the underlying task. Inherently interpretable models, on the other hand, circumvent these issues by explicitly encoding explanations into model architecture, meaning their explanations are naturally faithful, but they often exhibit poor predictive performance due to their limited expressive power. In this work, we identify a key reason for the lack of faithfulness of feature attributions: the lack of robustness of the underlying black-box models, especially to the erasure of unimportant distractor features in the input. To address this issue, we propose *Distractor Erasure Tuning* (`DiET`), a method that adapts black-box models to be robust to distractor erasure, thus providing discriminative and faithful attributions. This strategy naturally combines the ease of use of post hoc explanations with the faithfulness of inherently interpretable models. We perform extensive experiments on semi-synthetic and real-world datasets, and show that `DiET` produces models that (1) closely approximate the original black-box models they are intended to explain, and (2) yield explanations that match approximate ground truths available by construction. Our code is made public here.

## 1 Introduction

An important desideratum of machine learning models is for their predictions to be explainable. This allows both human domain experts as well as laypeople to better understand and trust the decisions made by models, and furthermore, is also a regulatory requirement for high-stakes settings. For example, both the European General Data Protection Regulation (GDPR) [1] and the US AI Bill of Rights [2], require organizations to provide explanations for decisions made in high-stakes settings. A common approach to producing such explanations from black-box models in a post hoc manner is via feature attribution, which aims to identify important input features influencing a model prediction. These methods typically work by locally approximating non-linear models with linear ones [3] under some input perturbations such as feature erasure. Intuitively, if the underlying model is more sensitive

37th Conference on Neural Information Processing Systems (NeurIPS 2023).

to the erasure of feature A than feature B, these methods aim to attribute a higher "importance" to feature A than feature B. A fundamental prerequisite for a feature to be considered important by a model is that it must first be useful in predicting the label, that is, it must be discriminative for the task. If a feature does not contain information relating to the output label, then it cannot be used to predict the label, and thus feature attribution methods must not consider them important. However, recent works [4, 5] have found this not to be case – feature attribution methods often highlight non-discriminative features. This motivates a natural question: what causes feature attributions to highlight such non-discriminative features, making them unfaithful?

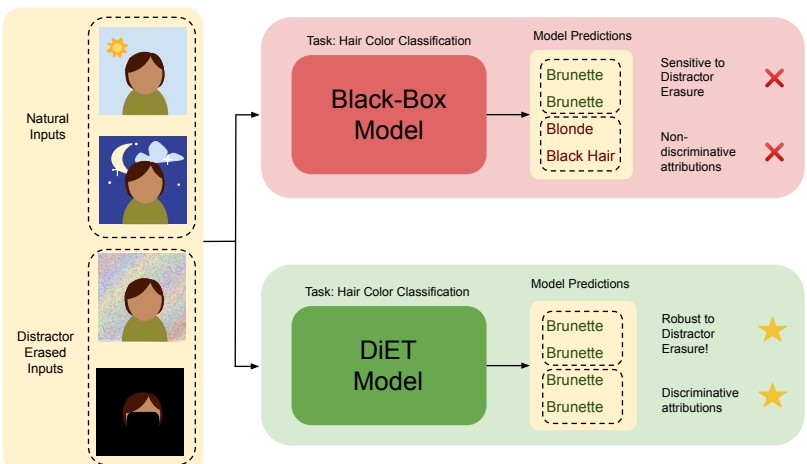

Figure 1: Illustration of our method, Distractor Erasure Tuning. `DiET` models exhibit robustness to distractor erasure (non-discriminative features such as backgrounds), allowing for the recovery of discriminative attributions.

Answering this question has been hard because of a lack of theoretical understanding of the faithfulness of feature attributions. While these notions have been used empirically [4, 5] to assess the quality of attributions, the theoretical characterization of optimally faithful feature attributions is missing in the literature. In this work, we tackle this problem by proposing a framework for feature attribution methods emphasizing faithfulness and particularly discriminability, formalized via the *signal-distractor decomposition* for datasets. Essentially, the signal denotes the discriminative parts of the input (relative to a given task), while the distractor denotes the unimportant parts. Feature attribution methods are then evaluated on how well they are able to recover the signal, thus also providing a well-defined notion of a "ground-truth". We theoretically identify an important criterion to recover this ground-truth, that being the robustness of the model to the erasure of the input distractors. To enable black-box models to recover such ground truth attributions, we propose *Distractor Erasure Tuning* (`DiET`), a method that adapts models to be robust to the erasure of input distractors. Given that these distractor regions are not known in advance, our method works by alternating feature attribution and model adaptation. At a high level, our work still uses feature attribution methods while adapting black-box models to have faithful attributions. Thus, this strategy naturally combines the ease of use of post hoc explanation methods with the faithfulness benefits of inherently interpretable models by providing the best of both alternatives. Our contributions are the following:

1. We present a formalism for feature attribution that emphasizes discriminability and allows for a notion of well-defined ground truth, via the signal-distractor decomposition of a dataset. We show that it is necessary for models to be robust to distractor erasure for them to be able to recover this ground truth.

2. We propose *distractor erasure tuning* (`DiET`), a novel method that adapts black-box models to make them robust to distractor erasure.

3. We perform experiments on semi-synthetic and real-world computer vision datasets and show that `DiET` outputs models that are faithful to its original input models, and that their feature attributions are interpretable and match ground-truth feature attributions.

## 2 Related Work

**Post-Hoc Explainability.** Post-hoc explainability methods aim to explain the outputs of fully trained black-box models either on an instance-level or global level. The most common post-hoc methods are feature attribution methods that rank the relative importance of features, either by explicitly producing perturbations [6, 7], or by computing variations of input gradients [8–10]. Perturbation-based methods are especially popular in computer vision literature [11–14], which use feature removal strategies adapted specifically for image data. However, these methods all assume a specific form for feature removal, and we show theoretically in Section 3 that this can lead to unverifiable attributions.

**Inherently Interpretable Models.** Inherently interpretable models are constructed such that we know exactly what they do, either through their weights or explicit modular reasoning. As such, the explanations provided by these models are more accurate than those given by post-hoc methods; however, the performance of interpretable models often suffers when compared to unconstrained black-box architectures. The most common inherently interpretable model classes include linear models, decision trees and rules with limited depth, GLMs, GAMs [15], JAMs [16–18], prototype- and concept-based models [19, 20], and weight-input aligned models [21]. While [19, 20] leverage the expressivity of deep networks, they constrain hypothesis classes significantly and still often suffer from a decrease in performance. Among these, our work most closely relates to JAMs, which amortise feature attribution generation using a learnt masking function to generate attributions in a single forward pass, and trains black-box models using input dropout. On other hand, JAMs (1) trains models from scratch, whereas `DiET` can interpret black-box models, (2) amortises feature attributions using a masking function resulting in less accurate attributions, (3) trains models to be robust to a large set of candidate masks via input dropout, leading to low predictive accuracy, whereas `DiET` trains models only to be robust to the optimal mask, leading to more flexibility and higher predictive accuracy.

**Evaluating Correctness of Explanations.** As explainability methods grow in number, so does the need for rigorous evaluation of each method. Research has shown that humans naively trust explanations regardless of their "correctness" [22], especially when explanations confirm biases or look visually appealing. Common approaches to evaluate explanation correctness rely on feature / pixel perturbation [23, 9, 24], i.e., an explanation is correct if perturbing unimportant feature results in no change of model outputs, whereas perturbing important features results in large model output change. Hooker et al. [5] proposed remove-and-retrain (ROAR) for evaluating feature attribution methods by training surrogate models on subsets of features denoted un/important by an attribution method and found that most gradient-based methods are no better than random. While prior works focused on developing metrics to evaluate correctness of explanations, our method `DiET` produces models that have explanations that are accurate by design, according to pixel-perturbation methods.

## 3 Theory of Discriminative Feature Attributions

In this section, we provide a theoretical framework for feature attribution, including a well-defined ground truth. We start by identifying a common feature of feature attributions, their reliance on perturbations or erasure. Intuitively, feature attribution methods work by simulating removing certain features and estimating how the model behaves when those features are removed: removing unimportant features should not change model behaviour. Typically, this erasure is implemented by replacing features with scalar values, such as the mean of the dataset [11, 12]. However, this can result in out-of-distribution inputs that can confuse a classifier, thus making it difficult to create meaningful attributions.

To ground this argument in an example, consider a model that classifies cows and camels. For an image of a camel, a feature attribution might note that only the hump of the camel and the sand it stands on are important for classification. As such, we would expect that the sky was irrelevant to the classifier's prediction, and we can concretely test this by altering it and creating a counterfactual sample. For example, we could mask the sky with an arbitrary uniform color; however, this may result in the sample being out-of-distribution for the model, and its prediction may change drastically even if the sky was not important for prediction. There are two strategies to overcome this problem. The first solution involves masking the sky in a manner that preserves the naturalness of the image,

but this solution involves using large-scale generative models, which themselves can contain biases and be uninterpretable. The second solution requires the classifier to be invariant to the erasure of the pixels corresponding to the sky, which is our solution in this paper. We formalize this argument below by defining an erasure-based feature attribution method called $(\epsilon, \mathcal{Q})$-feature attribution.

**Notation.** Throughout this paper, we shall assume the task of classification with inputs $\mathbf{x} \sim \mathcal{X}$ with $\mathbf{x} \in \mathbb{R}^d$ and $y \in [1, 2, ...C]$ with $C$-classes. We consider the class of deep neural networks $f : \mathbb{R}^d \to \triangle^C$ which map inputs $\mathbf{x}$ onto a $C$-class probability simplex. This paper considers binary feature attributions, which are represented as binary masks $\mathbf{m} \in \{0, 1\}^d$, where $\mathbf{m}_i = 1$ indicates an important feature and $\mathbf{m}_i = 0$ indicates an unimportant feature.

## 3.1 Feature Attributions with Input Erasure

We first define an erasure-based feature attribution such that the feature replacement method is explicit, and features are replaced with samples from a counterfactual distribution $\mathcal{Q}$. Particularly, we are interested in binary attributions (i.e., a feature is considered important or not) instead of real-valued ones.

**Definition 1.** $(\epsilon, \mathcal{Q})$-feature attribution (QFA) is a binary mask $\mathbf{m}(f, \mathbf{x}, \mathcal{Q})$ that relies on a model $f(\cdot)$, an instance $\mathbf{x}$, and a $d$-dimensional counterfactual distribution $\mathcal{Q}$, and is given by

$$\mathbf{m}(f, \mathbf{x}, \mathcal{Q}) = \arg\min_{\mathbf{m}'} \|\mathbf{m}'\|_0 \text{ such that } \mathbb{E}_{q \sim \mathcal{Q}} \|f(\mathbf{x}_s(\mathbf{m}', q)) - f(\mathbf{x})\|_1 \leq \epsilon$$

where $\mathbf{x}_s(\mathbf{m}, q) = \mathbf{m} \odot \mathbf{x} + (1 - \mathbf{m}) \odot q$

Thus, an $(\epsilon, \mathcal{Q})$-feature attribution (henceforth, *QFA*) refers to the sparsest mask that can be applied to an image such that the model's output remains approximately unchanged. QFA depends on the feature replacement distribution $\mathcal{Q}$, where $\mathcal{Q}$ is independent of both $\mathbf{x}$ and $y$. This generalizes the commonly used heuristics of replacing unimportant features with the dataset mean, in which case $\mathcal{Q}$ is a Dirac delta distribution at the mean value. The choice of $\mathcal{Q}$ is indeed critical, as an incorrect choice can hurt our ability to recover the correct attributions due to the resulting inputs being out-of-distribution and the classifier being sensitive to such changes. Specifically, an incorrect $\mathcal{Q}$ can result in QFA being less sparse, as masking even a few features with the wrong $\mathcal{Q}$ would likely cause large deviations in the model's outputs. As a result, given a model, we must aim to find the $\mathcal{Q}$ that leads to the sparsest QFA masks. However, the problem of searching over $\mathcal{Q}$ is complex, as it requires searching over the space of all $d$-dimensional distributions, and furthermore, if the underlying model is non-robust, there may not exist any $\mathcal{Q}$ that leads to sparse attributions. To avoid this, we consider the inverse problem: given $\mathcal{Q}$, we find the class of models that have the sparsest QFAs w.r.t. that particular $\mathcal{Q}$. We call this the $\mathcal{Q}$-robust model class, which we define below:

**Definition 2.** $\mathcal{Q}$-robust model class $\mathcal{F}_v(\mathcal{Q})$: For some given distribution $\mathcal{Q}$, the class of models $\mathcal{F}_v$ for which $\mathcal{Q}$ has the sparsest QFA mask as opposed to any other $\mathcal{Q}'$, such that for all $f \in \mathcal{F}_v(\mathcal{Q})$,

$$\mathcal{Q} = \arg\min_{\mathcal{Q}'} \mathbb{E}_{\mathbf{x}} \|\mathbf{m}(f, \mathbf{x}, \mathcal{Q}')\|_0$$

Intuitively, $\mathcal{Q}$-robust models result in the sparsest QFA masks and can be thought of as being robust to the erasure of "irrelevant" input features. Recalling our example of the cows and camels, we would like models to be robust to the replacement of the pixels corresponding to the sky but not necessarily robust to pixels corresponding to the camel or the cow itself. This distinguishes it from classical robustness definitions, which require models to be robust to small perturbations (rather than erasure) at every feature uniformly. Thus $\mathcal{Q}$-robustness is equivalent to enforcing robustness to the erasure of distractor features, a notion that is central to this work. For the rest of this paper, we shall refer to QFA applied to a model from a $\mathcal{Q}$-robust model class as a "matched" feature attribution – the same $\mathcal{Q}$ is used to both define the model class and the feature attribution.

## 3.2 Recovering the Signal-Distractor Decomposition

In the study of feature attribution, the 'ground truth' attributions are often unspecified. Here, we show that for datasets that are signal-distractor decomposable, formally defined below, there exists a ground truth attribution, and feature attributions for optimal verifiable models are able to recover it.

Intuitively, given an object classification dataset between cows and camels, the "signal" refers to the regions in the image that are discriminative, or correlated with the label, such as the cows or camels. The distractor refers to everything else, such as the background or sky. Note that if objects in the background are spuriously correlated with the label, i.e. sand or grass, those would be part of the signal, not the distractor. We first begin by formally defining the signal-distractor decomposition.

**Definition 3.** *A labelled dataset $D = \{(\mathbf{x}, y)_{i=1}^N\}$ has a signal-distractor decomposition defined by masks $\mathbf{m}(\mathbf{x}) \in \{0,1\}^d$ for every input $\mathbf{x}$, where*

1. *$\mathbf{x} \odot \mathbf{m}(\mathbf{x})$ is the discriminative signal, where $p(y \mid \mathbf{x}) = p(y \mid \mathbf{x} \odot \mathbf{m}(\mathbf{x}))$*

2. *$\mathbf{x} \odot (1 - \mathbf{m}(\mathbf{x}))$ is the non-discriminative distractor, where $p(y \mid \mathbf{x} \odot (1 - \mathbf{m}(\mathbf{x}))) = p(y)$*

3. *$\mathbf{m}(\mathbf{x})$ is the sparsest mask, i.e., $\mathbf{m}(\mathbf{x}) = \arg\min_{\mathbf{m}'(\mathbf{x})} \|\mathbf{m}'(\mathbf{x})\|_0$, such that (1) and (2) are satisfied.*

We propose to use the masks $\mathbf{m}(\mathbf{x})$ implied by the signal-distractor decomposition as ground truth feature attributions. These are meaningful as they precisely highlight the discriminative components of the image and ignore the non-discriminative regions. Discriminability has previously been considered an important criterion to evaluate feature attributions [4, 5] however, we here take a step further and propose its usage as ground truth.

We observe first that the masks $\mathbf{m}(\mathbf{x})$ of the signal-distractor decomposition always exist: setting $\mathbf{m}(\mathbf{x})$ as an all-ones vector trivially satisfies conditions (1) and (2). When multiple masks exist, condition (3) requires us to choose the sparsest such mask $\mathbf{m}(\mathbf{x})$. Using the definitions provided, we show below an asymptotic argument stating that $Q$-robustness is a necessary condition to recover the optimal masks defined by the signal-distractor decomposition.

**Remark**: A dataset $\mathcal{D}$ is said to have a "non-redundant signal" when the sparsest mask in condition (3) of the signal-distractor decomposition is equal to the sparsest mask when (1) alone is satisfied.

**Theorem 1.** *QFA applied to $Q$-robust models recover the ground-truth masks when applied to the Bayes optimal predictor $f_v^* \in \mathcal{F}_v(Q)$, for datasets $\mathcal{D}$ with a non-redundant signal.*

*Proof Idea.* We first note that the optimal $Q$ for QFA is equal to the ground truth distractor distribution, as this leads to the sparsest QFA. If a $Q$-robust model aims to recover the sparsest masks, then its QFA mask must equal that obtained by setting $Q$ equal to the distractor. From the uniqueness argument in the definition of the signal-distractor decomposition, this is possible only when the optimal mask is recovered by QFA. $\square$

**Corollary.** *QFA fails to recover the ground-truth masks when applied to predictors $f \notin \mathcal{F}_v(Q)$.*

This follows from the fact that for any $f \notin \mathcal{F}_v(Q)$, there exists some other $Q'$ that results in a sparser mask, indicating that the ground truth masks are not recovered. Thus, this shows that feature attributions applied to the incorrect model class can be less effective - in this case, they fail to recover the ground truth masks. Further, the Bayes optimality is an important condition because it ensures that the resulting model is sensitive to all discriminative features in the input – sub-optimal models are sub-optimal precisely because they fail to capture the signal from all the discriminative components of the input, and this can interfere with such models being able to recover ground truth masks. In practice, if we expect our models to be highly performant, we can expect them to be sensitive to all discriminative parts of the input and thus approximately recover ground truth masks. Finally, in practice, we do not have access to the ground truth masks for natural datasets, as the discriminative and the non-discriminative regions are not known in advance. In order to use these notions of ground truth, it is thus vital to construct semi-synthetic datasets where the discriminative parts are known. Thus, one can use semi-synthetic datasets to validate a feature attribution approach and then apply it to gain insight into real datasets with unknown signal and distractor components.

To summarize, we have defined a feature attribution method with the feature removal process made explicit via the counterfactual distribution $Q$. To minimize the sparsity of the attribution masks with a given $Q$, we use models from the $Q$-robust model class $\mathcal{F}_v(Q)$. Finally, we find that feature attributions derived from Bayes optimal models in the model class $\mathcal{F}_v(Q)$ are able to recover the ground-truth masks and fail to do so otherwise.

## 4 DiET: Distractor Erasure Tuning

In the previous section we showed that given a $\mathcal{Q}$-robust model $f_v \in \mathcal{F}_v(\mathcal{Q})$, we are able to apply QFA to recover the ground truth masks. In this section, we shall discuss how to practically build such robust models, given a pre-defined counterfactual distribution $\mathcal{Q}$ that defines the erasure method.

**Relaxing QFA.** We note that QFA as defined in definition 1 is difficult to optimize in its current form due to its use of $\ell_0$ regularization and its constrained form. To alleviate this problem, we perform two relaxations: first, we relax the $\ell_0$ objective into an $\ell_1$ objective, and second, we convert the constrained objective to an unconstrained one by using a Lagrangian. The resulting objective function is given in equation 1. Assuming the model $f_v$ is known to us, we can minimize this objective function to obtain $(\epsilon, \mathcal{Q})$-feature attributions for each point $\mathbf{x} \in \mathcal{X}$.

$$\mathcal{L}_{\text{QFA}}(\theta, \{\mathbf{m}(\mathbf{x})\}_{\mathbf{x} \in \mathcal{X}}) = \mathbb{E}_{\mathbf{x} \in \mathcal{X}} \left[ \underbrace{\|\mathbf{m}(\mathbf{x})\|_1}_{\text{mask sparsity}} + \lambda_1 \underbrace{\|f_v(\mathbf{x}; \theta) - f_v(\mathbf{x}_\text{s}(\mathbf{m}, q)); \theta)\|_1}_{\text{data distillation}} \right] \quad (1)$$

**Enforcing Model Robustness via Distillation.** Assuming that the optimal masks denoting the signal-distractor decomposition are known w.r.t. every training data point (i.e., $\{\mathbf{m}(\mathbf{x})\}_{\mathbf{x} \in \mathcal{X}}$), one can project any black-box model into a $\mathcal{Q}$-robust model via distillation. Specifically, we can use equation 2 for this purpose, which contains (1) a data distillation term to enforce the $\epsilon$ constraint in QFA, and (2) a model distillation term to enforce that the resulting model and original model are approximately equal. Accordingly, the black-box model $f_b$ and our resulting model $f_v$ both have the same model architecture, and we initialize $f_v = f_b$.

$$\mathcal{L}_{\text{train}}(\theta, \{\mathbf{m}(\mathbf{x})\}_{\mathbf{x} \in \mathcal{X}}) = \mathbb{E}_{\mathbf{x} \in \mathcal{X}} \left[ \underbrace{\|f_v(\mathbf{x}; \theta) - f_v(\mathbf{x}_\text{s}(\mathbf{m}(\mathbf{x}), q)); \theta)\|_1}_{\text{data distillation}} + \lambda_2 \underbrace{\|f_b(\mathbf{x}) - f_v(\mathbf{x}; \theta)\|_1}_{\text{model distillation}} \right] \quad (2)$$

**Alternating Minimization between $\theta$ and $\mathbf{m}$.** We are interested in both of the above objectives: we would like to recover the optimal masks from the dataset, as well as use those masks to enforce $(\epsilon, \mathcal{Q})$ constraints via distillation to yield our $\mathcal{Q}$-robust models. We can thus formulate the overall optimization problem as the sum of these terms, as shown in equation 3. Notice that both these objectives assume that either the optimal masks, or the robust model is known, and in practice, we know neither. A common strategy in cases that involve optimizing over multiple sets of variables is to employ alternating minimization [25], which involves repeatedly fixing one of the variables and optimizing the other. We handle the constrained objective on the mask variables via projection, i.e., using hard-thresholding / rounding to yield binary masks.

$$\theta^*, \{\mathbf{m}^*(\mathbf{x})\} = \arg\min_{\theta, \mathbf{m}} \left( \mathcal{L}_{\text{train}}(\theta, \{\mathbf{m}(\mathbf{x})\}) + \mathcal{L}_{\text{QFA}}(\theta, \{\mathbf{m}(\mathbf{x})\}) \right) \quad (3)$$

$$\text{such that} \quad \mathbf{m}(\mathbf{x}) \in \{0, 1\}^d \quad \forall \mathbf{x} \in \mathcal{X}$$

**Iterative Mask Rounding with Increasing Sparsity.** In practice, mask rounding makes gradient-based optimization unstable due to sudden jumps in the variables induced by rounding. This problem commonly arises when dealing with sparsity constraints. To alleviate this problem, use a heuristic that is common in the model pruning literature [26] called iterative pruning, which involves introducing a rounding schedule, where the sparsity of the mask is gradually increased during optimization steps. Inspired by this choice, we employ a similar strategy over our mask variables.

**Practical Details.** We implement these objectives as follows. First, we initialize the robust model to be the same as the original model, $f_v = f_b$, and the mask to be all ones, $D_s = m \odot D_d, m = 1$. We then iteratively (1) simplify $D_s$ by optimizing $\mathcal{L}_{QFA}$ until $\mathbf{m}$ converges, (2) round $\mathbf{m}$ such that it is binary (i.e. $D_s$ is a subset of features in $D_d$ rather than a weighting of them), and (3) update $f_v$ by minimizing $\mathcal{L}_{train}$ such that $D_s$ is equally as informative as $D_d$ to $f_v$ and $f_v$ is functionally equivalent

to $f_b$. As per Definition 2, we replace masked pixels in $D_s$ with a pre-determined counterfactual distribution $\mathcal{Q}$. This ensures that the given $\mathcal{Q}$ is the optimal counterfactual distribution for $f_v$, meaning $f_v$ comes from the $\mathcal{Q}$-robust model class $\mathcal{F}_v(\mathcal{Q})$. The pseudocode is given in Algorithm 1.

---

**Algorithm 1** Distractor Erasure Tuning

---

**Input:** Dataset $D_d := (x, y)$, model $f_b$, hyperparameter $k$ rounding steps
**Hyperparameters:** $k$ rounding steps, $u$ mask scaling factor, $s(t)$ sparsity at step $t$
$\{\mathbf{m}(\mathbf{x})\}$, s.t. $\mathbf{m}(\mathbf{x}) \leftarrow$ ones with shape $\mathbb{R}^{d/u}$ ▷ Init mask $\mathbf{m}(\mathbf{x})$ to ones
$f_v \leftarrow f_b$ ▷ Init robust model $f_v$ to $f_b$
**for** $k$ rounding steps **do**
    **while** $\mathcal{L}_{\text{QFA}}$ not converged **do**
        $\{\mathbf{m}(\mathbf{x})\} \leftarrow \{\mathbf{m}(\mathbf{x})\} + \nabla_{\mathbf{m}}\mathcal{L}_{\text{QFA}}$
    **end while**
    $\mathbf{m} \leftarrow \text{round}(\mathbf{m}, s(t)) \quad \forall\, \mathbf{m} \in \{\mathbf{m}(\mathbf{x})\}$
    **while** $\mathcal{L}_{\text{train}}$ not converged **do**
        $f_v \leftarrow f_v + \nabla_{\theta}\mathcal{L}_{\text{train}}$
    **end while**
**end for**
**return** $\{\mathbf{m}(\mathbf{x})\}, f_v$

---

**Mask Scale.** In order to encourage greater "human interpretability," we increase the pixel size of the masks by lowering their resolution. We do this by downscaling the masks before optimization. Concretely, we initialize the masks to be of size $m_d = x_d/u$, where $x_d$ is the dimension of the image $\mathbf{x}$ and $u$ is the pixel size we wish to consider. We then upsample the mask to be of dimension $x_d$ before applying it to $\mathbf{x}$. The more we downscale the mask by (i.e. the greater $u$ is), the more interpretable and visually cohesive the distilled dataset $\mathbf{x}_s$ is.

## 5 Experimental Evaluation

In this section, we present our empirical evaluation in detail. We consider various quantitative and qualitative metrics to evaluate the correctness of feature attributions given by DiET models as well as the faithfulness of DiET models to the models they are meant to explain. We also evaluate DiET models' ability to explain models manipulated to have arbitrary uninformative input gradients. Finally, we analyze the effect of the mask downscaling hyperparameter on attributions. Comparisons to additional baselines beyond those shown in Figure 2 are given in 8.

**Datasets.** **Hard MNIST:** The first is a harder variant of MNIST where the digit is randomly placed on a small subpatch of a colored background. Each sample also contains small distractor digits and random noise patches. For this dataset, we consider the signal to be all pixels contained within the large actual digit, and the distractor to be all pixels in the background, noise, and smaller digits. **Chest X-ray:** Second, we consider a semi-synthetic chest x-ray dataset for pneumonia classification [27]. To control exactly what information the model leverages such that we can create ground truth signal-distractor decompositions, we inject a spurious correlation into this dataset. We randomly place a small, barely perceptible noise patch on each image in the "normal" class. We confirm that the model relies only on the spurious signal during classification by testing the model's drop in performance when flipping the correlation (adding the noise patches to the "pneumonia" class) and seeing that the accuracy goes from 100% to 0%. As such, for this dataset, the signal is simply the noise patch and the distractor is the rest of the xray. **CelebA:** The last dataset is a subset of CelebA [28] for hair color classification with classes {dark hair, blonde hair, gray hair}. We correlate the dark hair class with glasses to allow for qualitative evaluation of each methods' ability to recover spurious correlations. This dataset does not have a ground truth signal distractor decomposition, as there are many unknown discriminative spurious correlations the model may rely upon.

**Models.** For all experiments, we use ImageNet pre-trained ResNet18s for our baseline models, $f_b$. All models achieve over 96% test accuracy. We train DiET models with $Q \sim \mathbb{1}_{d*d} * \mathcal{N}(\mu(D_d), \sigma^2(D_d))$, meaning that each image is masked with a uniform color drawn from a normal distribution around the mean color of the dataset. For all evaluation, we use $Q \sim \mathbb{1}_{d*d} * \mathcal{N}(\mu(D_d), 0)$

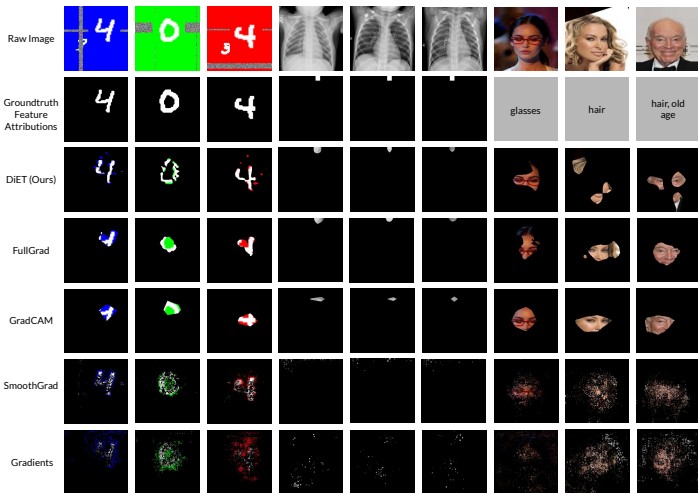

Figure 2: Visualization of datasets and attribution methods considered in this work. *Row 1*: Raw data samples, *Row 2*: Ground truth attributions, *Row 3*: `DiET` attributions, *Rows 4-7*: Baseline methods.

(the dirac delta of the dataset mean) to ensure that masked samples are minimally out-of-distribution for the baseline models $f_b$.

## 5.1 Evaluating the Correctness of Feature Attributions

**Pixel Perturbation Tests.** We test the faithfulness of our explanations via the pixel perturbation variant proposed in [9, 5], where we mask the $k$ least salient pixels as determined by any given attribution method and check for degradation in model performance with the mean of the dataset. This metric evaluates whether the $k$ masked pixels were necessary for the model to make its prediction. As mentioned in previous works, masked samples come from a different distribution than the original samples, meaning poor performance after pixel perturbation can either be a product of the model's performance on the masks or the feature attribution scores being incorrect. To disentangle the correctness of the attributions from the robustness of the model, we perform pixel perturbation tests on ground truth feature attributions, with results reported in the appendix. Note that our method returns binary masks, but this metric requires continuous valued attributions to create rankings. As such, for this experiment we use the attributions created by our method *before rounding*.

Results are shown in Figure 3. We find that the attributions produced by `DiET`, used in conjunction with `DiET` models outperform all baselines. Furthermore, `DiET` attributions tested on the baseline model also generally perform better than gradient-based attributions.

**Intersection Over Union.** We further evaluate the correctness of our attributions by measuring their Intersection Over Union (IOU) with the "ground truth" attributions. We use the signal from the ground truth signal-distractor decomposition as described in 5 for the ground truth attributions. For each image, if the ground truth attribution is comprised of $n$ pixels, we take the intersection over union of the top $n$ pixels returned by the explanation method and the $n$ ground truth pixels, meaning an IOU of 1 corresponds to a perfectly aligned/correct attribution. Results are shown in 1, where our method performs the best for both datasets. We report mean and standard deviation over the dataset for each method.

## 5.2 Evaluating the Faithfulness of `DiET` Models

To ensure that the `DiET` model ($f_v$) returned by our method is faithful the the original model ($f_b$) it approximates, we test the accuracy of `DiET` models with respect to the predictions produced by the original model. Specifically, we take the predictions of the original model $f_b$ to be the labels of the dataset. We evaluate `DiET` models on both the original dataset ($f_v(D_d) \approx f_b(D_d)$) and the simplified dataset ($f_v(D_s) \approx f_b(D_d)$), with results shown in 2. We see that `DiET`models are highly faithful to the baseline models they approximate.

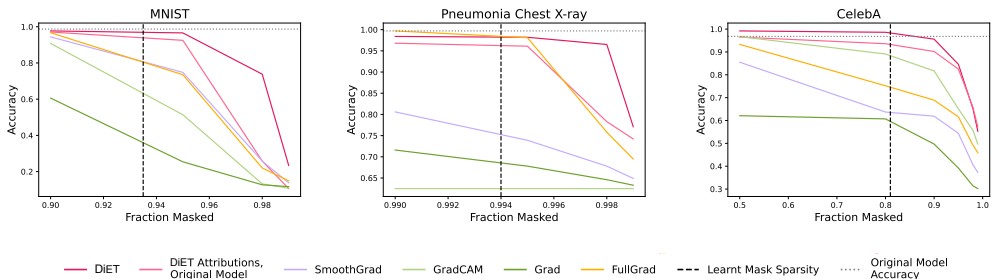

Figure 3: Pixel perturbation tests (higher is better) for MNIST (left), Chest X-ray (middle), and CelebA (right) datasets. `DiET`'s recommended mask sparsity is shown as a vertical dashed line. We observe that `DiET` performs the best overall. Refer to Section 5.1 for details.

Table 1: Intersection Over Union Results

|  | Hard MNIST | Chest X-ray |
| --- | --- | --- |
| `DiET` (Ours) | **0.461** ±0.08 | **0.821** ± 0.05 |
| SmoothGrad | 0.252±0.05 | 0.045 ±0.05 |
| GradCAM | 0.295 ±0.09 | 0.000 ±0.00 |
| Input Grad | 0.117 ±0.05 | 0.017±0.03 |
| FullGrad | 0.389 ±0.10 | 0.528 ±0.12 |

Table 2: Faithfulness of `DiET` to Original Model

|  | Hard MNIST | Chest X-ray | CelebA |
| --- | --- | --- | --- |
| Faithfulness on Original Data | 0.996 | 1.00 | 0.995 |
| Faithfulness on Simplified Data | 0.987 | 1.00 | 0.975 |

## 5.3 Qualitative Analysis of Simplified Datasets

We first explore how well `DiET` recovers signal-distractor decompositions. For CelebA, we see that all methods except for input gradients correctly recover the spurious correlation for the "dark hair/glasses" class, however only our method provides useful insights into the other two classes. We see that our method correctly identifies hair as the signal for the "blonde hair" class, whereas other methods simply look at the eyes, which are not discriminative. Furthermore, we see that for the "gray hair" class, our method picks up on hair, as well as initially unknown spurious correlations such as wrinkles and bowties. For Hard MNIST, we see that our method clearly isolates the signal and ignores the distractor. FullGrad and GradCAM suffer from a locality bias and tend to highlight the center of each digit. SmoothGrad and vanilla gradients are much noisier and highlight edges and many random background pixels. For the Chest X-ray dataset, we see that our method and FullGrad perfectly highlight the spurious signal. GradCAM again suffers from a centrality bias, and cannot highlight pixels on the edge. SmoothGrad and gradients appear mostly random to the human eye.

We also consider the visual quality of our attributions compared with the baselines (examples are shown in 2). We find that our method, FullGrad, and GradCAM appear the most interpretable, as opposed to SmoothGrad and vanilla gradients, because they consider features at the superpixel level rather than individual pixels. We also see that GradCam and FullGrad seem relatively class invariant, and tend to focus on the center of most images, rather than the discriminative features for each class, providing for less useful insights into the models and datasets.

## 5.4 Robustness to Adversarial Manipulation of Explanations

In this section, we highlight our method's robustness to adversarial explanation manipulation. To this end, we follow the manipulation proposed in [29], which adversarially manipulates gradient-based explanations. This is achieved by adding an extra term to the training objective that encourages input gradients to equal an arbitrary uninformative mask of pixels in the top left corner of each image. Note that model accuracy on the classification task is the same as training with only cross-entropy loss.

We repeat experiments for all evaluation metrics on these manipulated models, with pixel-perturbation shown below 6, and IOU, model faithfulness, and model robustness in the appendix. We see that gradient-based methods perform significantly worse on manipulated models; however, our method

remains relatively invariant. We also note while the models are only manipulated to have arbitrary input gradients, SmoothGrad and GradCAM are also heavily affected such that their attributions are entirely uninterpretable as well, as shown below.

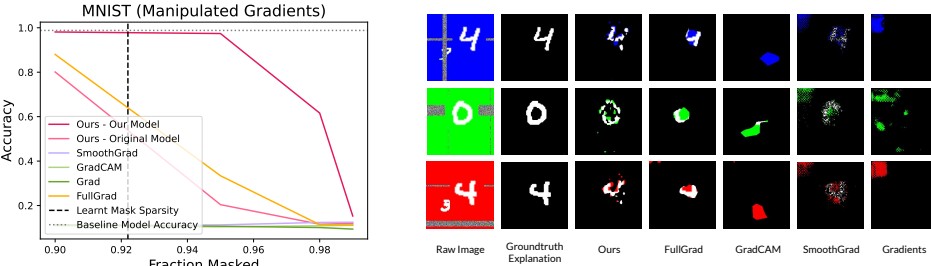

Figure 4: Pixel perturbation test and example images for models trained with gradient manipulation on Hard MNIST. Results for other datasets are in the appendix. Refer to Section 5.4 for details.

## 5.5 Attribution Sensitivity to Hyperparamaters

We conduct an ablation study on the choice of how much to downscale the mask by. The less we downscale by, the more fine-grained the mask is, allowing for optimization over a larger set of masks. However, the more we downscale by, the visually cohesive or "interpretable" to humans the masks are. We evaluate the trade-off between these two via pixel perturbation tests over multiple downscaling factors and with qualitative evaluations of the final masks in 5. We see that a downscaling factor of 8 performs the best on pixel perturbation tests. Increased factors of downscaling impose a greater locality constraint that results in informative pixels being masked, as shown in the visualization.

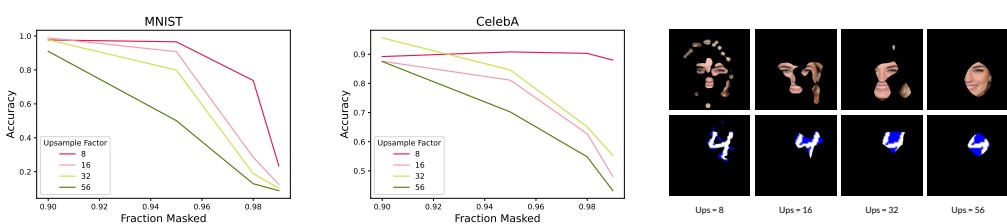

Figure 5: Ablation study of pixel perturbation test with varying levels of mask downscaling for MNIST and CelebA. Example images for each factor are shown on the right.

## 6 Discussion

In this paper, we seek to build machine learning models such that their feature attributions remain discriminative. In particular, we propose `DiET`, a method that adapts black-box models into those that are robust to distractor erasure. We empirically evaluate `DiET` and find that the resulting models are highly faithful to the original and produce interpretable attributions that closely match the ground truth ones. Limitations of `DiET` include requiring full access to the training dataset and the baseline model. Furthermore, while it produces verifiable feature attributions that tell us how important each feature is to the model's prediction, it does not tell us what the relationship between important features and the output/label is, as is true with all feature attributions.

## Acknowledgments and Disclosure of Funding

This work is supported in part by the NSF awards IIS-2008461, IIS-2040989, IIS-2238714, Kempner Institute Graduate Fellowship, and research awards from Google, JP Morgan, Amazon, Harvard Data Science Initiative, and the Digital, Data, and Design (D$^3$) Institute at Harvard. The views expressed here are those of the authors and do not reflect the official policy or position of the funding agencies.

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

## Supplementary Material

## 7 Proofs

**Theorem 2.** *$\mathcal{Q}$-feature attributions applied to $\mathcal{Q}$-robust models $(\mathcal{F}_v(\mathcal{Q}))$ recover the signal-distractor decomposition for Bayes-optimal predictor $f_v^* \in \mathcal{F}_v(\mathcal{Q})$.*

*Proof.* Consider QFA with $\epsilon = 0$. Let $f^*$ be the Bayes-optimal model, which implies that $f_y^*(\mathbf{x}) = p_{bayes}(y \mid \mathbf{x}) = p(y \mid \mathbf{x}) = \frac{p(\mathbf{x}|y)}{\sum_i p(\mathbf{x}|y=i)}$, i.e., the model estimates the correct conditional probabilities from the data given the class-conditional generative probabilities. Note that this is only defined for inputs $\mathbf{x} \in \mathcal{X}$ in the support of the data distribution and not outside, i.e., $p_{bayes}(y \mid \mathbf{x}) = p(y \mid \mathbf{x})$ only when $\mathbf{x} \in \mathcal{X}$.

Let us define $\mathbf{x} \odot (1 - \mathbf{m}) = \mathbf{x}_{\text{distractor}} \sim \mathcal{X}_{\text{distractor}}$. This is the distribution of distractor images, which (recall) are independent of the label $y$. Using this, we consider an **idealized version of QFA**, with $\mathcal{Q}_{ideal} = \mathcal{X}_{\text{distractor}}$. From Definition 1, this results in generation of simplified inputs $\mathbf{x}_s = \mathbf{x} \odot \mathbf{m} + q \odot (1 - \mathbf{m})$. For $q \sim \mathcal{Q}_{ideal} = \mathcal{X}_{\text{distractor}}$, we observe that $\mathbf{x}_s \in \mathcal{X}$, the data distribution. Recall the Bayes optimal classifier is defined across the data distribution $\mathcal{X}$, and applying QFA results in the sparsest $\mathbf{m}$ such that $p(y \mid \mathbf{x}) = p_{bayes}(y \mid \mathbf{x}) = p_{bayes}(y \mid \mathbf{x}_s) = p(y \mid \mathbf{x} \odot \mathbf{m} + q \odot (1 - \mathbf{m})) = p(y \mid \mathbf{x} \odot \mathbf{m})$. The last equality holds because $q \odot (1 - \mathbf{m})$ is independent of $y$. This corresponds to the definition of the signal-distractor, and thus it implies that **QFA with $\mathcal{Q}_{ideal}$ recovers the mask defined by the signal-distractor decomposition**.

For any other value of $\mathcal{Q} \neq \mathcal{Q}_{ideal}$, we first consider the $\mathcal{Q}$-robust Bayes optimal predictor $p_{bayes}^{\mathcal{Q}}(y \mid \mathbf{x})$. This has the property that $p_{bayes}^{\mathcal{Q}}(y \mid \mathbf{x}) = p_{bayes}(y \mid \mathbf{x})$ for $\mathbf{x} \in \mathcal{X}$. We now compare mask $\mathbf{m}_{\mathcal{Q}}$ derived from applying QFA on $p_{bayes}^{\mathcal{Q}}$ and mask $\mathbf{m}_{\mathcal{Q}_{ideal}}$ from applying QFA with $\mathcal{Q}_{ideal}$ on $p_{bayes}^{\mathcal{Q}}$. From the previous paragraph, we know that $\mathbf{m}_{\mathcal{Q}_{ideal}} = \mathbf{m}_{ideal}$ is the ideal sparsest mask. However from Definition 2, $\mathbf{m}_{\mathcal{Q}}$ is the sparsest mask. Thus it is the case that $\mathbf{m}_{\mathcal{Q}} = \mathbf{m}_{ideal}$, proving our overall result. $\square$

We now present proof for an additional statement not described in the main text, where we connect QFA to other commonly used feature attributions via the local function approximation framework [3] as follows.

**Theorem 3.** *QFA is an instance of the local function approximation framework (LFA), with (1) random binary perturbations, and (2) an interpretable model class consisting of linear models with binary weights*

*Proof.* Assume a black box model given by $f_b(\mathbf{x}; \mathbf{m}) = \mathbb{1}\left(\mathbb{E}_q \|f(\mathbf{x}_s(\mathbf{m}, q)) - f(\mathbf{x})\|_2 \leq \epsilon\right)$, loss function $\ell(f, g, x, \xi) = (f(x; \xi) - g(\xi))^2$, neighborhood perturbation $Z = \text{Uniform}(0, 1)^d$, and an interpretable model family $\mathcal{G}$ being the class of binary linear models.

For these choices, it is easy to see that

$$\arg\min_{g \in \mathcal{G}} \ell(f, g, \mathbf{x}, \xi)$$

$$= \arg\min_{g \in \mathcal{G}} \mathbb{E}_\xi \left(f_b(\mathbf{x}; \xi) - g^\top \xi\right)^2 + \lambda \|g\|_0$$

$$= \arg\min_{g \in \mathcal{G}} \mathbb{E}_\xi \left(\mathbb{1}\left(\mathbb{E}_q \|f(\mathbf{x}_s(\xi, q)) - f(\mathbf{x})\|_2 \leq \epsilon\right) - g^\top \xi\right)^2 + \lambda \|g\|_0$$

This above objective is minimized when $g = \mathbf{m}^*$, i.e., the ideal $\epsilon \mathcal{Q}$-FA mask, because this sets the first term to be zero by definition, and the second sparsity term ensures the minimality of the mask. $\square$

# 8 Additional Results

**Model Verifiability.** We further test the verifiability of our models by evaluating how the model's performance changes when performing the pixel perturbation test on groundtruth attributions. This enables us to disentangle the verifiability of the model from the correctness of the attributions, as we know that our attributions are correct. We use the same groundtruth attributions as in 5.1. We report the $\ell_1$ norm between predictions made on the original samples and predictions made on the masked samples. We compare our verifiable models to the baseline models they approximate, as well as models trained with input dropout, which [18] proposes as their verifiable model class. Training with input dropout is equivalent to training $f_v$ with random masks and cross-entropy loss rather than optimized masks and $f_b$ prediction matching. Results are shown in 3, where we see that our model performs similarly for masked and normal samples, whereas the other models do not.

|  | Hard MNIST | Chest X-ray |
|---|---|---|
| `DiET` Model $(f_v)$ | **0.027** | **0.0009** |
| Original Model $(f_b)$ | 0.107 | 0.0032 |
| $f_b$ + Input Dropout | 0.167 | 0.0536 |

Table 3: Model Verifiability

**Robustness to Explanation Attacks.** We report additional results on Chest X-ray and CelebA for the pixel perturbation tests, IOU tests, and model faithfulness tests for baseline models trained with manipulated gradients, as outlined in 5.4. We see that `DiET`models are still highly faithful and produce correct explanations even when derived from models adversarially trained to have manipulated explanations.

Table 4: Faithfulness of `DiET`Model for Manipulated Models

|  | Hard MNIST | Chest X-ray | CelebA |
|---|---|---|---|
| Accuracy on Original Data $f_v(D_d) = f_b(D_d)$ | 0.990 | 1.00 | 0.970 |
| Accuracy on Simplified Data $f_v(D_s) = f_b(D_d)$ | 0.980 | 1.00 | 0.946 |

Table 5: Intersection Over Union Results for Manipulated Models

|  |  | MNIST (manipulated) | Chest X-ray (manipulated) |
|---|---|---|---|
| Method | `DiET`(Ours) | **0.454** $\pm 0.08$ | **0.631**$\pm 0.12$ |
|  | SmoothGrad | 0.158 $\pm 0.07$ | 0.000 $\pm 0.00$ |
|  | GradCAM | 0.040$\pm 0.06$ | 0.000 $\pm 0.00$ |
|  | Input Grad | 0.002$\pm 0.01$ | 0.000 $\pm 0.00$ |
|  | FullGrad | 0.333 $\pm 0.12$ | 0.004 $\pm 0.04$ |

**Signal vs Distractor Masking.** For our CelebA experiments, there may have been unintended spurious correlations that we did not foresee and that our method did not recover, leading to a vacuous decomposition. To further support the results in Figure 3, we also perform experiments on masking the signal instead of the distractor, by masking out "important" pixels as opposed to the unimportant ones. The results for this experiment are shown in 6.

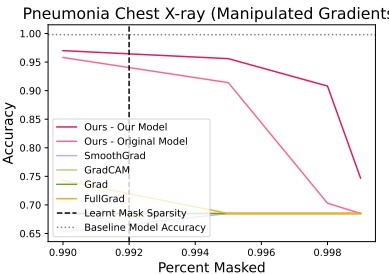 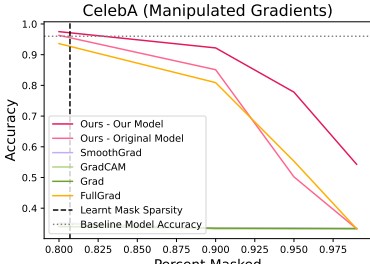

Figure 6: Pixel perturbation tests for models trained with gradient manipulation for Chest X-ray (middle) and CelebA (right).

Table 6: Signal Masking experiments

| Fraction Masked | Masked Distractor Acc | Masked Signal Acc | Random Mask Acc |
|---|---|---|---|
| 0.2 | 0.971 | 0.772 | 0.965 |
| 0.4 | 0.967 | 0.569 | 0.935 |
| 0.6 | 0.967 | 0.392 | 0.826 |
| 0.8 | 0.960 | 0.339 | 0.587 |

These results indicate that for the `DiET` models, there do not exist pixels outside the shown signal regions in Figure 2 that contain information about the label. These results show that the signal-distractor decomposition for CelebA is very likely non-vacuous.

**Finetuning on less data.** We recognize that a limitation of this work is that it requires further training of the given models on their training datasets. To address this issue, we briefly explore whether `DiET` can be finetuned on a small subset of the data distribution (without labels) as follows for MNIST. The original model is trained on 8835 samples from the train split. `DiET` is finetuned on 1500 samples from a separate unlabeled validation split. We perform pixel perturbation on the remaining 8500 test samples. We compare this to `DiET` models finetuned on the full train set and note a minimal change in performance in 7.

Table 7: Small Scale Finetuning

| Percent Masked | `DiET`(Finetuned on Train Split) | `DiET`(Finetuned on Val Split) |
|---|---|---|
| 90 | 0.977 | 0.962 |
| 95 | 0.966 | 0.937 |
| 98 | 0.738 | 0.691 |
| 99 | 0.234 | 0.288 |

Overall, this shows that there exist ways to decrease the computational complexity of our procedure when applied to larger datasets, which we haven't fully investigated yet.

**Effect on standard robustness.** We conduct additional robustness experiments for Gaussian and Bernoulli noise with varying standard deviations and probabilities. Results are shown in 8 and 9. We find that `DiET` models are generally more robust than regular models. Note that neither `DiET` models nor the original models are explicitly trained or tuned for robustness on these distributions.

**Additional Baselines: SHAP and BCosNets.** We consider two additional explanation baselines: SHAP [7] and BCosNets [21]. Results for pixel perturbation tests and visualizations are shown in 7. Results for IOU tests are shown in 10.

Table 8: Robustness experiments (Gaussian noise)

|      | Hard MNIST | | CelebA | |
| STD | Ours | Original | Ours | Original |
| --- | --- | --- | --- | --- |
| 0.2 | 0.974 | 0.976 | 0.959 | 0.939 |
| 0.4 | 0.961 | 0.892 | 0.675 | 0.537 |
| 1.0 | 0.61 | 0.176 | 0.335 | 0.438 |

Table 9: Robustness experiments (Bernoulli noise)

|      | Hard MNIST | | CelebA | |
| p | Ours | Original | Ours | Original |
| --- | --- | --- | --- | --- |
| 0.9 | 0.182 | 0.155 | 0.348 | 0.356 |
| 0.75 | 0.718 | 0.3 | 0.393 | 0.364 |
| 0.5 | 0.922 | 0.748 | 0.528 | 0.453 |
| 0.1 | 0.962 | 0.969 | 0.867 | 0.807 |

We note that in general SHAP does not often perform as well as most gradient-based methods for image data. We find that it performs similarly to random attributions.

We add B-CosNets as an inherently-interpretable model baseline. We find that B-CosNets performs comparably to our method for the IOU test on Hard MNIST. We were unable to train it to convergence on the Chest X-ray dataset. We also find that visualizations created by B-CosNets align with our expectations and appear visually interpretable. However, our method still significantly outperforms B-CosNets on the pixel perturbation test, showing that B-CosNets are not robust to perturbations of the distractor (i.e. are not verifiable). We also note that our method can be applied to a trained black-box model of any architecture and training procedure, whereas B-CosNets, like all inherently interpretable models, cannot.

Table 10: **SHAP and B-Cos IOU.** Note that B-CosNets did not converge for the Chest X-ray dataset.

|      | Hard MNIST | Chest X-ray |
| --- | --- | --- |
| `DiET` | $0.461 \pm 0.08$ | $0.821 \pm 0.05$ |
| B-CosNets | $0.465 \pm 0.07$ | – |
| SHAP | $0.036 \pm 0.02$ | $0.016 \pm 0.05$ |

**Effect of Choice of Q.** We perform an ablation to test the effect that the choice of Q has empirically. We consider various parameterizations of the normal distribution used for Q, and find that results are relatively consistent across the different choices of Q, as shown in 8.

# 9 Additional Implementation and Computation Details

Models were trained on the original train/test split given by `https://github.com/jayaneetha/colorized-MNIST` for Hard MNIST and [27] for the Chest X-ray dataset and with a random 80/20 split for CelebA. Baseline models were trained with Adam for 10 epochs with learning rate $1e-4$ and batch size 256. All hyperparameters are included in the code for this paper. The model distillation and data distillation terms are weighted with $\lambda_1 = \lambda_2 = 1$. We learn our masks with SGD (lr=300, batch size = 128) and our robust models with Adam (lr=$1e-4$, batch size = 128). We ran all experiments on a single A100 80 GB GPU with 32 GB memory.

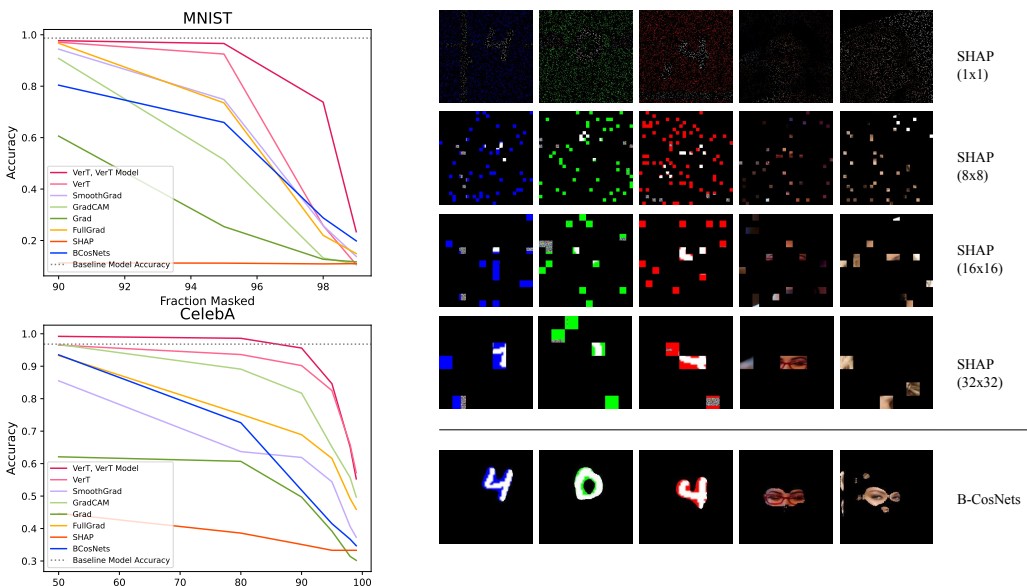

Figure 7: **Left:** Pixel perturbation tests with added baselines of SHAP (orange) and B-CosNets (blue). **Right:** Visualization of SHAP and B-CosNets explanations on Hard MNIST and CelebA. Results for SHAP are shown at varying feature sizes. We note that SHAP explanations appear relatively random at all levels of granularity.

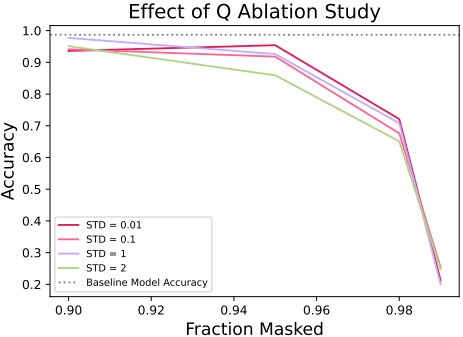

Figure 8: Pixel perturbation tests on Hard MNIST for `DiET` models trained with various $Q$s, $Q \sim \mathcal{N}(\mu, \lambda\sigma)$, for $\lambda$ in $\{0.01, 0.1, 1, 2\}$.

## 10 Broader Impact

Our method, `DiET`, aims to transform black-box models into distractor robust, interpretable models, which produce easily verifiable feature attributions. As such, if applied correctly, it can help users and stakeholders of machine learning models better understand a model's predictions and behavior by isolating the features necessary for each prediction, which can help highlight biases, overfitting, mistakes, and more. It can also help to identify spurious correlations that naturally exist in datasets and are leveraged by models through identification of the signal-distractor decomposition. However, even if `DiET`does not identify a spurious correlation, that does not mean that further dataset cleaning, processing, or curation is not needed, as a different model may still learn a spurious correlation that was not leveraged by the original model. Furthermore, feature attributions often do not constitute a

*complete* explanation of a model. For instance, while an attribution tells us *what* was important, it does not tell us *how* it was important or how the model uses that feature. In all high stakes applications, it is still imperative that stakeholders think critically about each prediction and explanation, rather than blindly trusting either.

