# OpenReview forum: "Discriminative Feature Attributions: Bridging Post Hoc Explainability and Inherent Interpretability"
_NeurIPS.cc/2023/Conference — NeurIPS 2023 poster_

### Official Review · Reviewer_dxrg · 2023-06-10

**Soundness:** 3 good
**Presentation:** 4 excellent
**Contribution:** 3 good
**Rating:** 7
**Confidence:** 4

**Summary:**

This paper introduces VerT, a method to distill *verifiable* models from black-box models. Concretely, These verifiable models are distilled by fitting a model $f_v$ that reproduces the predictions of the black-box model $f_b$ on a training set. Unlike the black-box model, the model $f_v$ is made verifiable by making its predictions self-consistent if a mask isolating the signal is applied to the input data. This mask is fitted by adding an extra objective to the model distillation objective, which guarantees that the mask is sparse and that the predictions of $f_v$ is similar for both the masked and the unmasked input. The methods is evaluated on 3 image datasets, including 2 where ground-truth feature importance is known. This analysis demonstrates that VerT outperforms standard gradient-based feature importance methods.

**Strengths:**

**Good writing.** The paper was really easy to read and follow. The notations are on-point and intuitive.

**Solid empirical validation.** The experiments provided in Section 5 are convincing and thoroughly verify important claims: the features identified as salient by VerT have the strongest impact when masked (which is not surprising given the optimization objective underlying VerT), these features have a reasonable overlap with the ground-truth salient features in a setting where the later are known and VerT shows improved robustness. The authors also show that VerT improves the robustness to adversarial attacks on the explanations  with respect to gradient-based feature importance methods (which is unsurprising as the attacks are designed to fool gradient-based methods specifically). Insightful illustrative examples to understand the gains of VerT are provided in Figure 2.

**Weaknesses:**

**Restrictive assumption on replacement distribution.** In Lines 125-137, the authors discuss the importance of the choice of the replacement distribution $\mathcal{Q}$ in order to avoid creating OOD examples by masking. It should be mentioned that some important masking stategies, such as [Gaussian blurs](https://arxiv.org/abs/1704.03296), are omitted from this discussion as these are conditioned on the input image $\mathbf{x}$. I am wondering if it is even possible to define masking strategies with replacement distributions *independent* of the input $\mathbf{x}$ that do not create OOD examples. Intuitivelly, it is legitimate to expect the replacement input $\mathbf{q}$ to have (at least) some information about $\mathbf{x}$ to avoid replacing the features of $\mathbf{x}$ by OOD values. I would recommend the author to discuss this point thoroughly and (possibly) acknowledge this as a limitation of the work.

**Unrealistic signal-distractor decomposition.** The signal-distractor decomposition defined in Definition 3 is key for Theorem 1. It appears to me that this decomposition is unrealistic for a simple reason: in the underlying DGP, the mask $\mathbf{m}$ and the signal $\mathbf{s}$ are independent. To make this point more clear, I would like to consider the example discussed in the paper. If we consider a cow detection task, the signal $\mathbf{s}$  would typically correspond to the cow-part of the image. If that is the case, the mask $\mathbf{m}$ should depend on the position of the cow on the image (e.g. performing a translation on the signal $\mathbf{s}$ should result in a similar translation on the mask $\mathbf{m}$ in principle). Again, I would recommend the authors to comment on the realism of their assumptions.



### Minor Weaknesses

- Algorithm1: what is $M(\mathbf{x})$? Is this the same as $\mathbf{m}(\mathbf{x})$? Also, should there be a minus sign in front of the gradients if the objectives are minimized?
- In the appendices, Theorem 2 corresponds to Theorem 1 the main paper.

**Questions:**

All my questions and recommendations are contained in the weakness section. I will not repeat them here to avoid redundancies. I will consider improving my recommendation if the above weaknesses are addressed by the authors.


**Limitations:**

Some limitations of the work are discussed in Section 6 of the main paper. I do not believe that negative societal impacts are a real concern for this work.

---

> ### Author Rebuttal · Authors · 2023-08-09
>
> Thank you for your constructive review!
>
>
> *“Restrictive assumption on replacement distribution. In Lines 125-137, the authors discuss the importance of the choice of the replacement distribution Q in order to avoid creating OOD examples by masking. It should be mentioned that some important masking stategies, such as Gaussian blurs, are omitted from this discussion as these are conditioned on the input image x. I am wondering if it is even possible to define masking strategies with replacement distributions independent of the input x that do not create OOD examples. Intuitivelly, it is legitimate to expect the replacement input q to have (at least) some information about x to avoid replacing features of x by OOD values. I would recommend the author to discuss this point thoroughly and (possibly) acknowledge this as a limitation of the work.“*
>
> We apologize for the confusion. Please take a look at our comment (G1) regarding this issue. While it is true that any such masking creates OOD examples, the critical issue here is that we can create models that are robust to such replacement of the unimportant features. In other words, we would like to modify models (via VerT) to be not sensitive to such OOD examples, where pixels are replaced with $Q$ in the non-important features. By predefining our feature replacement distribution $Q$, we can modify our models to be robust to these perturbed OOD examples while still maintaining that the replacement distribution is independent of the input $x$. We will clarify this in the paper.
>
> We do not discuss gaussian blur because in this case the “replacement distribution” $Q$ is dependent on $x$, however our framework requires them to be independent. For replacement distributions that are not independent of $x$, we are unable to disentangle the effect that the replacement distribution has on the model from the true signal in $x$, i.e., blurring does not completely remove the information in the image, but masking out the pixels does.
>
> ---
>
> *“Unrealistic signal-distractor decomposition. The signal-distractor decomposition defined in Definition 3 is key for Theorem 1. It appears to me that this decomposition is unrealistic for a simple reason: in the underlying DGP, the mask m and the signal s are independent. To make this point more clear, I would like to consider the example discussed in the paper. If we consider a cow detection task, the signal s would typically correspond to the cow-part of the image. If that is the case, the mask  m should depend on the position of the cow on the image (e.g. performing a translation on the signal  s should result in a similar translation on the mask m  in principle). Again, I would recommend the authors to comment on the realism of their assumptions.”*
>
> Great observation! This is a very subtle point about the signal-distractor decomposition. Note that definition 3 has redundant information: the mask being independently sampled from a distribution $m \sim \mathcal{M}$ implies $p(y | x) = p (y | s) = p(y | s \odot m)$ (because the distractor component $d \odot (1- m)$ is necessarily independent of the label), which is listed as an additional condition. Hence we can eliminate the $m \sim M$ statement and instead *define* mask $m$ to be a binary mask such that condition (1) $p(y | x) = p(y | s \odot m)$  and condition (2), that the mask $m$ is minimal, holds.
>
> From your example, this ensures that mask $m$ now depends on signal $s$, and a translation of the cow in the image results in a corresponding translation of the mask to satisfy conditions (1,2). This amendment to our definition does not impact Theorem 1 or the rest of our theory, and we hope this makes the resulting graphical model more realistic. We again thank the reviewer for noticing such a subtle issue.

---

> > ### Comment · Reviewer_dxrg · 2023-08-10
> >
> > I thank the authors for their thorough and clear rebuttal. I think that the explanation they provided about the signal-distractor decomposition is perfectly valid. I am still convinced of this work's quality in spite of the other reviews, and will therefore increase my score.

---

### Official Review · Reviewer_qGoL · 2023-06-20

**Soundness:** 2 fair
**Presentation:** 3 good
**Contribution:** 3 good
**Rating:** 5
**Confidence:** 4

**Summary:**

The present work introduces a theoretical framework of verifiability of feature attributions based on the sparest (binary) feature attribution mask that only barely changes the models' output. The authors theoretically (and empirically) show that for signal-distractor decomposable datasets off-the-shelf black-box models cannot be verified according to their definition of verifiability. The main reason being that off-the-shelf models cannot handle the OOD samples created by the masking intervention ("feature replacement"). To overcome this, they propose a finetuning scheme, in which they make off-the-shelf black-box models robust to such feature interventions. Specifically, they alternatingly optimize for the sparsest mask (that only barely changes the models' output) and apply a distillation loss. While the former makes the model robust to the feature interventions, the latter ensures the similarity to the original model. Experimentally, they show that VerT improves interpretability over gradient-based methods. VerT robustly identifies the signals (most salient features) of the inputs, while retaining the performance of the original model.

**Strengths:**

- The theoretical verifiability framework is simple yet sound.
- The theoretical analysis (Sec 3.2) is interesting and supported both theoretically as well as empirically.
- The finetuning scheme is simple yet effective. It does not change the models' output while simultaneously enhancing its interpretability by making it robust to the input feature removal.
- The paper is clearly written (except for some small issues; see questions & suggestions below) and easy to follow.
- Code is provided.

**Weaknesses:**

- The major weakness of the present work is that it does not compare to any removal-based feature attribution methods (e.g., [SHAP](https://proceedings.neurips.cc/paper_files/paper/2017/hash/8a20a8621978632d76c43dfd28b67767-Abstract.html), [FastShap](https://openreview.net/forum?id=Zq2G_VTV53T), etc.), models also trained on feature attributions (e.g., [right for the right reasons](https://www.ijcai.org/proceedings/2017/371)), nor inherently interpretable models (e.g., [JAM](https://arxiv.org/abs/2103.01890) or [B-Cos](https://openaccess.thecvf.com/content/CVPR2022/html/Bohle_B-Cos_Networks_Alignment_Is_All_We_Need_for_Interpretability_CVPR_2022_paper.html)); only comparisons to gradient-based methods are provided.
- The experiments always include manually-defined spurious correlations (to contain clear signals and distractors). While this provides empirical evidence for their theoretical framework, it would be meaningful to compare the method on “untouched”, more challenging datasets besides Celeb, e.g., ImageNet.
- The finetuning scheme may change the model’s behavior (but not its output due to Eq. 2). While Tab. 2 shows that the prediction stays similar, it may change what type of signals in the inputs the classifier considers for its prediction (and its inner workings). Consequently, the finetuned model may not be faithful to which signal it uses (or how it processes it) for its prediction.

**Questions:**

- In Fig. 2, do all methods use the same model (finetuned or not finetuned version)?
- Is there any explanation for the large difference for the non-finetuned and finetuned models using VerT in Fig. 4? Is the adversarial training objective also used during finetuning? More generally, would finetuning without model distillation (Eq. 2) and data distillation in Eq. 1 lead to more robust (and interpretable) models that only focus on the important parts of the image?
- How did the authors set $\lambda_1, \lambda_2$?

**Suggestions**
- [B-Cos networks](https://openaccess.thecvf.com/content/CVPR2022/html/Bohle_B-Cos_Networks_Alignment_Is_All_We_Need_for_Interpretability_CVPR_2022_paper.html) or [BagNets](https://openreview.net/forum?id=SkfMWhAqYQ) could also be mentioned as inherently interpretable models.
- The abbreviations in L68 could be introduced.
- There are several recent advancements of concept-based models that close the performance gap, e.g., [post-hoc CBMs](https://openreview.net/forum?id=HAMeOIRD_g9) that could be mentioned in the related work.
- Theorem 2 in the supplemental should have the same number as Theorem 1 in the main text as well as formulation to make it easier for readers to find.
- In the experiments section there are several missing “Table/Figure {num}”.
- The hyperparameters are neither described in the main text nor supplement.
- To which Figure/Table are L288-299 referring to?
- There are several typos throughout the present submission.
- Results for Celeb in Table 3 in the supplement are missing.

**Limitations:**

The limitations are adequately addressed.

---

> ### Author Rebuttal · Authors · 2023-08-09
>
> Thank you for your constructive review!
>
> *“The major weakness of the present work is that it does not compare to any removal-based feature attribution methods (e.g., SHAP, FastShap, etc.), models also trained on feature attributions (e.g., right for the right reasons), nor inherently interpretable models (e.g., JAM or B-Cos); only comparisons to gradient-based methods are provided.”*
>
> Please refer to our comment (G2) for discussion on differences with classic feature attributions (like SHAP). Nonetheless, we present comparisons with SHAP in comment (G5).
>
> Our paper differs from the paper “Right for the Right Reasons: Training Differentiable Models by Constraining their Explanations” (Ross et al., AAAI 2017) in that while their paper assumes (partial) knowledge of the ground truth feature attributions, our setup does not assume any such knowledge, thus making this incompatible with our setup.
>
> We have already presented a comparison with JAM in Table 3 in the Supplementary material, where JAMs are “f_b + input (pixel) dropout”, which is precisely the method proposed by JAMs to train interpretable models. We shall change that column to read “JAMs” instead.
>
> We have added B-CosNetsv2 (Bohle et al., 2023, "B-cos Alignment for Inherently Interpretable CNNs and Vision Transformers") as a baseline, with results shown in (G5). We find that they perform comparably for the IOU tests and qualitatively, however VerT still significantly outperforms for pixel perturbation and has the advantage of being applicable post-hoc. Thus, VerT models are verifiable, meaning that any attribution map has a precise meaning in terms of concrete perturbations made to the inputs via definition 1. On the other hand, B-CosNet models are not verifiable, meaning that perturbing non-important pixels leads to output change, thus making the attribution misleading wrt model behaviour, which is precisely what pixel perturbation shows.
>
> ---
>
> *“The experiments always include manually-defined spurious correlations (to contain clear signals and distractors). While this provides empirical evidence for their theoretical framework, it would be meaningful to compare the method on “untouched”, more challenging datasets besides Celeb, e.g., ImageNet.”*
>
> For natural datasets such as ImageNet, a popular evaluation for explanations includes the pixel perturbation test, which we already perform for our existing datasets. Note that this evaluation does not require the use of a ground truth feature attribution.
>
> ImageNet experiments also require considerably more computational resources and engineering to train models and store masks for every single training data point, and hence we are unable to do these during the rebuttal. However, we shall prioritize this experiment for a future version of this draft.
>
> ---
>
> *“The finetuning scheme may change the model’s behavior (but not its output due to Eq. 2). While Tab. 2 shows that the prediction stays similar, it may change what type of signals in the inputs the classifier considers for its prediction (and its inner workings). Consequently, the finetuned model may not be faithful to which signal it uses (or how it processes it) for its prediction.”*
>
> While this is an intriguing hypothesis, note that we are unable to either confirm or deny this, as it is fundamentally unclear what features are used by black-box models! This motivates our framework which emphasizes verifiability.
>
> Further, we also think any such change is irrelevant – since the Q-verifiable models are equivalent to the original, why not use the verifiable models instead, since they come with the additional benefit of interpretability?
>
> ---
>
> *“In Fig. 2, do all methods use the same model (finetuned or not finetuned version)?”*
>
> In Fig. 2, row 3 (VerT ours) uses the VerT model $f_v$ finetuned via our method to explain $f_b$. All other rows use the model, $f_b$, which is not finetuned via our objective. Note that $f_v$ and $f_b$ are functionally equivalent.
>
> ---
>
> *“Is there any explanation for the large difference for the non-finetuned and finetuned models using VerT in Fig. 4? Is the adversarial training objective also used during finetuning? More generally, would finetuning without model distillation (Eq. 2) and data distillation in Eq. 1 lead to more robust (and interpretable) models that only focus on the important parts of the image?”*
>
> Please refer to global comment (G1) for a clarification about VerT and connection to robustness. We do not use adversarial training for VerT. The objective of VerT training (model + data distillation) is precisely to make the models more mask-robust and hence interpretable. We are unsure if this answers your question, please let us know in case of any clarification.
>
> ---
>
> *"How did the authors set lambda_1, lambda_2?"*
>
> For all experiments, we set both $\lambda_1$ and $\lambda_2$ to 1. We will be sure to add this to our implementation details.
>
> Thank you for your suggestions regarding the writing! We will be sure to address these in the final draft.

---

> > ### Comment · Reviewer_qGoL · 2023-08-14
> > **Response to rebuttal**
> >
> > I thank the authors for their detailed response! Specifically, I appreciate the efforts to extend the experimental design (inclusion of SHAP & B-Cos). Nevertheless, I’d like to re-raise some concerns not sufficiently answered in my opinion.
> >
> > > “B-CosNet models are not verifiable, meaning that perturbing non-important pixels leads to output change, thus making the attribution misleading wrt model behaviour, which is precisely what pixel perturbation shows.”
> >
> > I have severe doubts that this claim is true since B-Cos networks are inherently interpretable and, consequently, automatically faithful & verifiable in a general sense; beyond the limited verifiability definition of Def. 1. Since they are inherently interpretable, the final statement (“thus making the attribution misleading wrt model behaviour”) is incorrect.
> >
> > This seems to also be related to my original concern that VerT may change the underlying model, while for B-Cos this is not the case. Thus, VerT may be over optimized for metrics like pixel-perturbation -as suggested by reviewer 2ozo- and is acknowledged by the authors (“But your observation that our method performs the best because it is robust to masking is correct”). On the other hand, B-Cos networks are not optimized and thereby may be susceptible to non-important pixels (due to limitations of the training procedure and data), which actually may be true, e.g., in the presence of spurious correlations (as mentioned by reviewer QRQ5) or similar. Thus, the attributions of such (obviously undesirable) non-important pixels faithfully contribute to the model’s output.
> >
> > > “however VerT still significantly outperforms for pixel perturbation and has the advantage of being applicable post-hoc.”
> >
> > The last statement (“[VerT] has the advantage of being applicable post-hoc”) feels like a contradiction since we need to finetune VerT first and cannot just apply it post-hoc. This is also acknowledged by the authors in the general response (“our method (VerT + QFA) involves adapting the model to the attribution method”).
> >
> > > “ImageNet experiments also require considerably more computational resources and engineering to train models and store masks for every single training data point, and hence we are unable to do these during the rebuttal.“
> >
> > While I acknowledge the honesty of the authors, this also highlights a major practical limitation to models trained on large-scale data.
> >
> > > “Another fundamental contribution of this work lies in its conceptual framework. Classical feature attribution literature does not have a precise notion of “ground truth”, making it fundamentally unclear what quantities these are estimating in the first place. In this paper, we introduce one notion of ground truth – the signal-distractor decomposition that is a property of the dataset itself (and NOT the model!).”
> >
> > I kindly refer the authors to previous works (e.g., [1]) that also used a signal-distractor framework.
> >
> > > “In particular, Q-verifiability can be thought of as a very specific form of robustness, where models are ONLY robust wrt the distractor in the input and NOT the signal, thus distinguishing it from classical robustness”
> >
> > Did the authors try to evaluate robustness on standard pipelines? It would be interesting to see if their proposed fine tuning scheme can go beyond mere input feature interpretability (my original comment had a typo…).
> >
> > ---
> >
> > [1] Kindermans, Pieter-Jan, et al. "Learning how to explain neural networks: Patternnet and patternattribution." ICLR 2018.

---

> > > ### Author Response · Authors · 2023-08-15
> > >
> > > We thank the reviewer for their response. We hope to clarify some points below.
> > >
> > > ---
> > >
> > > *I have severe doubts that this claim is true since B-Cosnets are inherently interpretable...*
> > >
> > > Our claim is that in our framework, if a part of the image is deemed not important, then our method ensures that perturbing those pixels do not change model outputs. This is critical to our notion of verifiability. Our assertion is that this doesn’t hold for Bcosnets, and the pixel perturbation experiments show this precisely.
> > >
> > > Overall, we believe that it is misleading for a pixel to be considered unimportant by an attribution method for a model, and simultaneously have random perturbations in that pixel leading to large changes in model outputs. This is precisely the scenario we aim to avoid with our method.
> > >
> > > Note that BCosNets have a different notion of “inherent interpretability” that does not involve perturbations and thus this is consistent with our earlier comment. We emphasize that while the attribution masks produced by BCosNets are nearly equivalent to ours in terms of the IOU with the ground truth, these attributions do not have a precise meaning in terms of model behavior on perturbations in their formalism.
> > >
> > > ---
> > >
> > > *VerT may change the underlying model, while for B-Cos this is not the case.*
> > >
> > > There is no “underlying black-box model” in B-CosNets, as these are trained from scratch. Furthermore, our method creates an “inherently interpretable” (verifiable) model that is trained to match the behavior of the original black-box model.
> > >
> > > ---
> > >
> > > *Thus, VerT may be over optimized for metrics like pixel-perturbation... On the other hand, B-Cosnets are not optimized and may be susceptible to non-important pixels, which actually may be true, e.g., in the presence of spurious correlations...*
> > >
> > > Please refer to our earlier response to your question about B-CosNets and inherent interpretability (the first response in this comment).
> > > Regarding spurious correlations, please also see our response to QRQ5 about the signal-distractor decomposition being in line with the spurious correlations.
> > >
> > > Also, the reviewer’s comment that “*thereby may be susceptible to non-important pixels (due to limitations of the training procedure and data), which actually may be true, e.g., in the presence of spurious correlations*” is incorrect and conflates pixel importance and spurious correlations. If a pixel is not important for a model and an input, then by definition it means that perturbing that pixel does not affect the model output. However, spurious correlations are a property of the dataset – if a spurious correlation is leveraged by a model (which we can verify experimentally), then pixels corresponding to these spurious correlations are “important” to the classifier!
> > >
> > > ---
> > >
> > > *The last statement (“[VerT] has the advantage of being applicable post-hoc”) feels like a contradiction...*
> > >
> > > We apologize for the confusion. By “post hoc'' in this case, we mean that VerT finetunes a black-box model, resulting in another model that closely approximates its behavior. However, Bcosnets train models from scratch, thus bearing no resemblance to any underlying black-box model. We shall avoid using this term to refer to this (important) distinction to avoid more confusion.
> > >
> > > ---
> > >
> > > *I kindly refer the authors to previous works...*
> > >
> > > Thanks for pointing us to that work! The provided reference formulates a signal-distractor decomposition only for linear models, whereas our formulation is more general and holds for non-linear models as well. We will cite this in our work, and also amend our claims to state that we are the first to formulate a general signal-distractor framework that applies to non-linear datasets and models.

---

> > > > ### Author Response · Authors · 2023-08-15
> > > >
> > > > ---
> > > >
> > > > *This also highlights a major practical limitation to models trained on large-scale data.*
> > > >
> > > > We briefly explore whether VerT can be finetuned on a small subset of the data distribution as follows for MNIST. The original model is trained on 8835 samples from the train split. VerT is finetuned on 1500 samples from a separate *unlabeled* validation split. We perform pixel perturbation on the remaining 8500 test samples. We compare this to VerT models finetuned on the full train set and note a minimal change in performance.
> > > >
> > > > | Percent Masked | VerT (Finetuned on Train Split) | VerT (Finetuned on Val Split) |
> > > > | --- | --- | --- |
> > > > | 90 | 0.977 | 0.962 |
> > > > | 95 | 0.966 | 0.937 |
> > > > | 98 | 0.738 | 0.691 |
> > > > | 99 | 0.234 | 0.288 |
> > > >
> > > > Overall, this shows that there exist ways to decrease the computational complexity of our procedure when applied to larger datasets, which we haven’t fully investigated yet.
> > > >
> > > > ---
> > > >
> > > > *Did the authors try to evaluate robustness on standard pipelines?*
> > > >
> > > > Thanks for suggesting the experiment! We conduct additional robustness experiments for Gaussian and Bernoulli noise with varying standard deviations and probabilities. We find that VerT models are generally more robust than regular models. Note that neither VerT models nor the original models are explicitly trained or tuned for robustness on these distributions.
> > > >
> > > > Gaussian:
> > > > |  | MNIST |  | CelebA |  |
> > > > | --- | --- | --- | --- | --- |
> > > > | STD | Ours | Original | Ours | Original |
> > > > | 0.2 | 0.974 | 0.976 | 0.959 | 0.939 |
> > > > | 0.4 | 0.961 | 0.892 | 0.675 | 0.537 |
> > > > | 1.0 | 0.61 | 0.176 | 0.335 | 0.438 |
> > > >
> > > > Bernoulli:
> > > > |  | MNIST |  | CelebA |  |
> > > > | --- | --- | --- | --- | --- |
> > > > | P | Ours | Original | Ours | Original |
> > > > | 0.9 | 0.182 | 0.155 | 0.348 | 0.356 |
> > > > | 0.75 | 0.718 | 0.3 | 0.393 | 0.364 |
> > > > | 0.5 | 0.922 | 0.748 | 0.528 | 0.453 |
> > > > | 0.1 | 0.962 | 0.969 | 0.867 | 0.807 |

---

### Official Review · Reviewer_2ozo · 2023-07-04

**Soundness:** 2 fair
**Presentation:** 2 fair
**Contribution:** 2 fair
**Rating:** 3
**Confidence:** 5

**Summary:**

This paper proposes a method called Verifiability Tuning (VerT), which transforms black-box models into models that naturally yield faithful and verifiable feature attributions. Authors further conduct experiments on semi-synthetic and real-world datasets to verify the effectiveness of the proposed VerT method.

**Strengths:**

1. The motivation of this paper is clear.

2. This paper focuses on the faithfulness of post hoc explanation methods, which is a very important topic in XAI.

**Weaknesses:**

1. The equation in Definition 1 is problematic. For example, let three positive input variables $a=b=c>0$ have a MAX operation, $output = \\max \\{a,b,c,0\\}$. Then, we mask any two input variables will result in different results. Specifically, we can mask $a$ and $b$, and keep $c$ unchanged. We can also mask $c$ and $b$, and keep $b$ unchanged. These two masking methods will result in different explanation results, but the actual importance of $a$, $b$, $c$ to the inference is the same. Hence, the equation in Definition 1 is problematic.

2. I disagree with your claim on the optimal Q. Specifically, the optimal Q can be found. Theoretically, the optimal Q should be the same as the distributions of the input image, i.e., setting q=x (in this case $\epsilon=0$ in Definition 1). Although this setting q=x conflicts with the motivation of masking the input, it is indeed the optimal solution to q in mathematics.

3. The ground truth of attribution methods constructed in experiments (in Figure 2) is too simple.

(1) If a classifier is powerful enough to just use five pixels for inference, instead of using the entire foreground of “4,” then is this classifier better or worse? The proposed evaluation does not consider this as a good explanation if annotating the entire foreground as the ground truth.

(2) Defining “4” as the foreground also seems problematic. The edge feature of “4”contains both pixels in foregrounds and pixels in background. Hence, information encoded in background can be used for inference theoretically. In this way, pixels outside of the “4” should also be considered as the ground truth. However, there exists another problem that how many pixels outside of“4”should be included as the ground truth.

(3) The correlation between dark hair and glass in CelebA dataset is just assumed. Specifically, this correlation is not a necessary condition for hair color classification, because the DNN can either exclusively use glasses for classification, or exclusively use hair, or use both hair and glasses for classification. The correlation is assumed and used for evaluation, but the correlation is not a certificated truth of a DNN.

This is a typical case when the input information for inference is redundant. When a small part of foreground objects are already enough for classification, it is difficult to annotate the ground-truth attention of a DNN.

4. Pixel perturbation tests are circular arguments, because the proposed method is learned by minimizing the loss. The loss is designed to mask unimportant pixels for inference.

5. Only the accuracy is not enough to evaluate the faithfulness. Please compare the advantages of the proposed method with advantages of the Shapley value. We cannot assume that the direct change of the output caused by the masking of an input variable is the exact importance of the input variable. Please apply more sophisticated evaluation metrics for attributions proposed in recent years.

6. Authors just compare the proposed method with gradient-based explanation method, which is a quite weak competing method. Authors are suggested to compare the proposed method with more sophisticated baselines, such as Shapley values, DeepLIFT, IG, etc.

7. Why “QFA applied to a model from a Q-verifiable model class” can be considered as “a verifiable feature attribution?” Authors do not provide proofs to support this claim. Moreover, what is the definition of “a verifiable feature attribution?” Please clarify how a feature attribution can be considered as “verifiable.”

**Questions:**

Please see weaknesses.

**Limitations:**

No, the authors do not discuss limitations of the proposed method.

---

> ### Author Rebuttal · Authors · 2023-08-09
>
> Thank you for your constructive review!
>
> *“The equation in Definition 1 is problematic. For example, let three positive input variables a=b=c ? 0 have a MAX operation, output= max{a,b,c,0} ...actual importance of a, b, c to the inference is the same...”*
>
> Great example! This shows that with duplicate features, the QFA optimization problem does not have a unique solution. This is expected, as there is no single “minimal” component of the input that drives the output, and this multiplicity highlights a critical property of the model.
>
> We’d also like to point out that similar notions of feature importance have been used in literature, see references [8,9,10] in the main paper. We’d like to further point out that defining “actual importance” of variables is an open problem, and there is no agreement in the field regarding what it constitutes (see reference [1] in our paper). Overall, we don’t see why multiplicity makes definition 1 problematic. Can you please elaborate in case we missed something?
>
> ---
>
> *“I disagree with your claim on the optimal Q. Theoretically, the optimal Q should be the same as the distributions of the input image, i.e., setting q=x (in this case e=0 in Definition 1)...”*
>
> This is incorrect. Please note that the distribution Q is independent of the instance x: samples from the same “Q” must be used for all inputs “x”. Thus it cannot be the case that q = x, where q $\sim$ Q.
>
> ---
>
> *“The ground truth of attribution methods constructed in experiments (in Figure 2) is too simple.”
> “(1) If a classifier is powerful enough...”
> “(2) Defining “4” as the foreground also seems problematic...”
> “This is a typical case when the input information for inference is redundant. When a small part of foreground objects are already enough for classification, it is difficult to annotate the ground-truth attention of a DNN.”*
>
> Good observation! For real-world datasets, a precise ground truth signal is fundamentally unknown, which is why we rely on semi-synthetic datasets where at least approximate information about the ground truth signal can be known. For hard-MNIST, while we also expected the classifier not to use all pixels of the digit “4” as the signal, VerT in Figure 2 seems to extract something very close – indicating that most of the pixels of “4” are indeed useful for classifying the digit. The purpose of our IOU experiments is to ensure that unimportant pixels (i.e. background pixels that are definitely not informative to the task) are not given more importance than pixels that may or may not be important (i.e. pixels that lie within the signal). All other baselines methods heavily attribute background pixels, whereas ours does not. We hope this addresses your question, and we are happy to clarify otherwise.
>
> ---
>
> *“(3) The correlation between dark hair and glass in CelebA dataset is just assumed..."*
>
> This is incorrect, we do not assume the correlation, rather we explicitly designed our dataset such that dark hair and glasses were correlated. All images of people with dark hair were chosen to have glasses, and all images with blonde or white hair do not have glasses. We further test that the model relies on this correlation by testing performance on dark haired people without glasses and blonde and white haired people with glasses, noting that accuracy drops from 97% to 38% (near guessing). We will clarify this in the paper.
>
> ---
>
> *“Pixel perturbation tests are circular arguments...”*
>
> We don’t completely agree – the Q we use to train the model is different from that used to perform the pixel perturbation method. But your observation that our method performs the best because it is robust to masking is correct, and it is precisely the point of our approach! Please see global comments (G1-G4). We think that this is an advantage of our method, as pixel perturbation is a commonly used test and aligns well with our intuitions for what feature attributions must satisfy. We present this evaluation as a sanity check with respect to known metrics for feature attribution.
>
> ---
>
> *“Only the accuracy is not enough to evaluate the faithfulness...”*
>
> Our core argument (G1) is that the feature perturbation strategy is critical. Typical implementations of Shapley values provide attributions of black-box models using an arbitrary feature perturbation strategy, whereas we propose to align the feature perturbation method with the model’s robustness to that perturbation method. We present a comparison with SHAP in (G5). We also present an extensive set of evaluations: pixel perturbation (Figure 3), IOU with an approximate ground truth (Table 1), robustness to explanation manipulation (Figure 4) and sensitivity to hyper-parameters (Figure 5). Is there a specific evaluation metric you had in mind that is conceptually different from these? If so, we would be happy to consider it.
>
> ---
>
> *“Authors just compare the proposed method with gradient-based explanation method...”*
>
> Please refer to (G2) for an explanation of our method and how it differs from classical feature attribution methods such as Shapley values, DeepLIFT, etc. Essentially, our method is orthogonal to these works. While most feature attribution methods attempt to perform feature attribution of black-box models, VerT aligns the model to the explanation method (via Q) used. We nonetheless present a comparison to SHAP in (G5) as requested.
>
> ---
>
> *“Why “QFA applied to a model from a Q-verifiable model class” can be considered as “a verifiable feature attribution?” Authors do not provide proofs to support this claim. Moreover, what is the definition of “a verifiable feature attribution?” Please clarify how a feature attribution can be considered as “verifiable.””*
>
> Please refer to (G1-G4) for clarification. We apologize for any confusion! That statement defines “verifiable feature attribution” in our paper, i.e., QFA applied to a model from a Q-verifiable model class. If you think this phrase is confusing, we are happy to consider renaming it.

---

> > ### Comment · Reviewer_2ozo · 2023-08-16
> > **Responses to authors**
> >
> > I would like to thank the authors for the detailed rebuttal. Some of my concerns are addressed (e.g., the lack of comparison between  the proposed method and advanced explanation methods, such as SHAP). However, the overall quality of the paper still does not meet the standard for publication, so I would keep my original rating. Nevertheless, it is good to hear that the authors are going to further polish the paper. I hope it can be presented in a clearer way and get accepted to a future conference.

---

> > > ### Author Response · Authors · 2023-08-16
> > >
> > > Thank you for your response! Can you please point to any specific concerns that remain unaddressed?

---

### Official Review · Reviewer_QRQ5 · 2023-07-07

**Soundness:** 2 fair
**Presentation:** 3 good
**Contribution:** 3 good
**Rating:** 5
**Confidence:** 4

**Summary:**

The paper proposes a way to verifiably get feature attributions of the ground truth signal when the input can be decomposed into independent signals and distractor features, assuming there is a counterfactual generator Q that can provide sparse attributions. The process consists of first deriving (\epsilon, Q) feature attributions for each point to get optimal masks, and second use distillation to train a new model that matches the outputs of the QFA with the original prediction while staying close to the original model. The first and second steps are alternated to minimize the overall loss. A rounding scheme is used to stabilize training with hard masks and superpixels are used. The approach is tested on MNIST, Chest X-ray, and CelebA.

Update: After the discussion, I have updated my score accordingly. Since the authors believe scenarios that satisfy the signal-distractor decomposition are likely to exist, it would be great to incorporate such a concrete example somewhere in the text.

**Strengths:**

The paper proposed an approach that, in principle, could potentially extract the ground truth signal (if all assumptions are met).

The approach, even though it trains a new model, replicates the original fairly well.

The experiments are couple with 2 ablations on adversarial robustness and sensitivity to hyperparameters.

**Weaknesses:**

The signal decomposition assumes that the signal and distractors are generated independently, and that therefore the correct feature attribution is the sparsest one. This is not going to always be the case in real settings.

The method requires a strong counterfactual generator Q to ensure correct recovery of sparse attributions. However, in practice the authors just use a dirac delta of the dataset mean which is not really a counterfactual generator.

The evaluation is specific to vision, and the two non-MNIST datasets have spurious correlations injected into the dataset. It'd be nice to have some kind of breadth here beyond vision, or to have some results on more natural data that did not need to be artificially correlated.

The claim that one can change any black box model into a verifiably interpretable model might be a bit strong---it seems to also depend on a number of assumptions (ground truth decomposability) and having a powerful generator in order for the "verified" part to hold true.

**Questions:**

Theorem 1 states that QFA applied to optimal predictors from the function class recovers the signal distribution. The corollary then states that QFA does not recover the signal when applied outside of the function class. Neither of these address non-optimal predictors from the function class---what happens there? The last sentence of the section seems to state that feature attributions from the function class are in fact able to recover the signal, but this does not seem to strictly follow from the theorem and corollary.

What superpixels are being used?

How do we know that we've reached the optimal f^*? It seems important to know this due to Theorem 1 requiring an optimal predictor.

The practical details states at the end that replacing masked pixels with a counterfactual distribution Q somehow ensures that Q is an optimal counterfactual distribution. It is then asserted that this means the f comes from the Q-verifiable model class. I am not following either of these logical jumps, can you explain? There is nothing about Q in the pseudocode either. This seems to be critical, since we need to find an optimal f from Q for the signal to be recovered. Is there an ablation you can use to show how much of an effect Q has experimentally?

It is not entirely clear what the simplified dataset is, which appears to be spontaneously used in Section 5.

Why is the evaluation focused on gradient-based feature attributions? Why not other classic local surrogates like LIME/SHAP which are also widely used?

**Limitations:**

The limitations do in fact mention that a decomposition must exist.

---

> ### Author Rebuttal · Authors · 2023-08-09
>
> Thank you for the constructive review! Overall, we feel you may have misunderstood some parts of the paper, and we’d like to clarify these below.
>
> *“The signal decomposition assumes that the signal and distractors are generated independently, and that therefore the correct feature attribution is the sparsest one. This is not going to always be the case in real settings.”*
>
> Please refer to global comments (G2, G4). Can you please provide examples of settings where you believe these may not hold?
>
> ---
>
> *“The method requires a strong counterfactual generator Q to ensure correct recovery of sparse attributions. However, in practice the authors just use a dirac delta of the dataset mean which is not really a counterfactual generator. ” “The claim that one can change any black box model into a verifiably interpretable model might be a bit strong”*
>
> Please refer to (G1, G3) for clarification. Theorem 1 states that **any** Q-verifiable model used with a corresponding QFA can recover the optimal signal-distractor distribution. Note that the Q here is unrelated to the distractor distribution ($\mathcal{X}_{distractor}$). The important thing here is that QFA is only used on Q-verifiable models with the same “Q” used for both – this is exactly what leads to verifiable explanations.
> Note also that for our method we use a Normal distribution for Q when training VerT models. We use the dirac delta of the dataset mean **only** for pixel perturbation evaluations to remain consistent with prior works and evaluation of baselines. We have also provided an ablation on the choice of Q in (G5).
>
> ---
>
> *“Theorem 1 states that QFA applied to optimal predictors from the function class recovers the signal distribution. The corollary then states that QFA does not recover the signal when applied outside of the function class. Neither of these address non-optimal predictors from the function class---what happens there? The last sentence of the section seems to state that feature attributions from the function class are in fact able to recover the signal, but this does not seem to strictly follow from the theorem and corollary. ” “How do we know that we've reached the optimal f^\*? It seems important to know this due to Theorem 1 requiring an optimal predictor.”*
>
> Please refer to (G3, G4). The main point is that QFA must be used with a Q-verifiable model class (with the same Q used for both) for recovery. When the predictor is non-optimal, intuitively the model may not learn to “look” at the right portions of the input, and thus may not recover the signal, which is a dataset-dependent quantity.
>
> Thanks for bringing to our notice the final line in the section, we will change it to “Finally, we find that feature attributions derived from **optimal models** in model class Fv (Q) are able to recover the signal-distractor decomposition of datasets.”
>
> ---
>
> *“What superpixels are being used?”*
>
> We do not use superpixels in this work. We instead use a low-resolution binary mask (say 8x8) which we upscale (to, say 224x224) using bilinear interpolation. Please refer to the paragraph on “Mask Scale” in line 222 of the paper. We use the word “superpixel” in that section loosely to refer to a group of pixels obtained from this procedure. We shall remove the usage of that word for clarity.
>
> ---
>
> *“The practical details states at the end that replacing masked pixels with a counterfactual distribution Q somehow ensures that Q is an optimal counterfactual distribution. It is then asserted that this means the f comes from the Q-verifiable model class. I am not following either of these logical jumps, can you explain? There is nothing about Q in the pseudocode either. This seems to be critical, since we need to find an optimal f from Q for the signal to be recovered. Is there an ablation you can use to show how much of an effect Q has experimentally? ”*
>
> We’re sorry for the confusion, please refer to (G1-G4). Our algorithm “verifiability tuning” converts a black-box model (which has an unknown Q) into a Q-verifiable model (for some Q), which results in the sparsest possible masks when used with QFA (with the same choice of Q).
>
> The results of our ablation are given in (G5), where we see that for different parameterizations of Q, the results are still relatively consistent.

---

> > ### Comment · Reviewer_QRQ5 · 2023-08-14
> >
> > Thanks for the response. I understand the claim about working for any Q and appreciate the additional results on SHAP.
> >
> > On decomposition: In the contrary, I would think that the signal-distractor decomposition is a very strong assumption that is rarely fulfilled in practice. Spurious correlations in the data is a well known problem in the data---i.e. cats/dogs and indoor/outdoor backgrounds, water environments in the waterbirds dataset, age and gender in CelebA, etc. Even benign correlations can fail to satisfy the assumption: any image photograph is affected by lighting, lens quality, corruptions, etc. which affects and correlates all pixels, and cannot be decomposed into two independent feature sets. As far as I'm aware, it is much rarer to have data that can always be split into two disjoint sets of features that are independently generated. After all, the examples in the submitted paper are either (a) synthetically created to satisfy the assumption or (b) do not satisfy the assumption (celebA).
> >
> > On optimal predictors: the response did not answer how we know we are using an optimal predictor, nor did it explain the logical jumps/pseudocode. I have looked at G1-G4 as the authors referred to but these did not answer the technical question. The former (how we know if the predictor is optimal) still appears to be a critical requirement.

---

> > > ### Author Response · Authors · 2023-08-15
> > >
> > > Thank you for your response! We hope to answer your questions regarding decomposition and optimal predictors.
> > >
> > > ---
> > >
> > > *On decomposition: In the contrary, I would think that the signal-distractor decomposition is a very strong assumption...*
> > >
> > > We stress that the notion of signal-distractor decomposition is perfectly in line with the existence of spurious correlations in the data! In fact, this was our motivation to begin with – given a dataset, we’d like to find the signal distractor decomposition, which is a property of the dataset that can be used to detect the presence of spurious correlations in the data. In other words, If the dataset has spurious correlations, then this is well-reflected in the signal portion of the signal-distractor decomposition. As an example, consider the waterbirds dataset, which has a spurious correlation between the label and the background. In this case, the signal also consists of the background (as this contains information about the label) in addition to the bird. Thus by inspecting the recovered signal distribution from a model, we are able to ascertain whether the dataset encodes spurious correlations.
> > >
> > > We agree with the reviewer that feature attribution methods (like ours) are unable to detect all sources of spurious correlations, and can in fact only detect those that can be localized in terms of individual pixels, and perhaps not those like “lighting, lens quality, corruptions, etc” which may be non-localized, like you mention. This has also been explored in other related works (Adebayo et al., “Post hoc Explanations may be Ineffective for Detecting Unknown Spurious Correlation”, ICLR 2022) which establishes that this failure case applies to ALL feature attribution methods. Our framework helps formalize these notions that precisely identify the technical limitations of using feature attribution. In other words, all datasets have a signal-distractor decomposition (i.e., some parts of the image may be completely uninformative to predicting the label), but this signal-distractor decomposition may not always help with diagnosing spurious correlations, especially when the spurious signal is non-localized.
> > >
> > > As a correction to the reviewer’s comment that “..do not satisfy the assumption (celebA)” (with the assumption being a signal-distractor decomposition), we note that even with CelebA, we selected a subset that contained a spurious correlation, such that “dark hair” and “glasses” were spuriously correlated, and we used this information to evaluate qualitatively whether feature attribution methods (including ours) are able to recover this spurious correlation. Note that this subset of CelebA does have a signal-distractor decomposition (like all datasets do), but it is just that in case we do not know it a priori.
> > >
> > > ---
> > >
> > > *On optimal predictors: the response did not answer how we know we are using an optimal predictor, nor did it explain the logical jumps/pseudocode. I have looked at G1-G4 as the authors referred to but these did not answer the technical question. The former (how we know if the predictor is optimal) still appears to be a critical requirement.*
> > >
> > >
> > > We apologize for the confusion.
> > >
> > > Regarding the jumps, are you referring to lines 219-221 in the paper? We mean the following: our procedure (VerT) aims to train models such that their QFA masks are as sparse as possible for a given choice of Q. This aligns with the definition of Q-verifiable models, as these are models with the sparest mask for that choice of Q.
> > >
> > > In practice, we use a Q that is a 3-dimensional Gaussian distribution, with mean equal to the dataset mean (which is a 3d RGB value for images), and a standard deviation of approximately 0.2 (which is the standard deviation of the dataset images). And this choice of Q is used for QFA in the code. However, this choice of Q is not critical, as we show in our experiments in the 1-page PDF. Please let us know if it is still unclear, we are happy to clarify further.
> > >
> > > Regarding optimal predictors, it is true that in practice we do not always know whether our predictors are optimal – Bayes optimal classifiers are a theoretical model that exist only when the complete data distributions are known in advance, i.e., $p(y \mid x) = p(x \mid y) p(y) / p(x)$.
> > >
> > > However, another way to check for optimality is to consider label noise. For instance, if the underlying dataset is clean and has no label noise, then a classifier that obtains 100% test accuracy is Bayes optimal. Real datasets always have small amounts of label noise, which upper bound their test accuracy (say ~99% for MNIST, ~96% for CIFAR, etc). In practice, if we have close-to-optimal models (i.e., models that perform well in terms of test accuracy), then in accordance with our theory, these must also approximately recover the signal-distractor decomposition, which we verify with our experiments.

---

> > > > ### Comment · Reviewer_QRQ5 · 2023-08-18
> > > > **On correlations**
> > > >
> > > > If I understand correctly, for the waterbirds example, you are saying that if backgrounds are correlated with the label, then it will be included in the signal? Wouldn't this mean that the entire input is considered signal, and so the decomposition is vacuous (i.e. there is no distractor subset)?
> > > >
> > > > I mentioned spurious correlations as an example but it doesn't really matter whether it is spurious or not (I also do not insist that the paper must detect spurious correlations). I was just wondering whether there is a realistic scenario where such a decomposition exists in practice, since most datasets are at least weakly correlated in all features (with spurious ones such as backgrounds being one of the most prominent examples from the literature). For example, I'm still not entirely sure that the chosen subset of CelebA has a nonvacuous decomposition. I understand that the chosen features of dark hair and glasses are purposefully correlated and so both are considered signals, but wouldn't these still be correlated with the other parts of the image and so there is no distractor component?
> > > >
> > > > As for the optimal predictors: Thank you for explaining. This makes more sense to me---I previously thought the "this" in the last sentence referred to only the immediately preceeding sentence and not the entire procedure. Would probably be good to make it less ambiguous in the revision.

---

> > > > > ### Author Response · Authors · 2023-08-21
> > > > >
> > > > > Thank you for your response! We address your questions below.
> > > > >
> > > > > *“If I understand correctly, for the waterbirds example, you are saying that if backgrounds are correlated with the label, then it will be included in the signal? Wouldn't this mean that the entire input is considered signal, and so the decomposition is vacuous (i.e. there is no distractor subset)?”*
> > > > >
> > > > > You are correct! For instances where all parts of the input statistically carry information about the label, the signal-distractor decomposition is indeed vacuous, and there is no distractor. Apriori, model designers collect a dataset with a specific informal prior of what an ideal
> > > > > signal-distractor decomposition for that task should look like. In the case of waterbirds, the prior might be that the background corresponds to the distractor. However, when this expectation is violated (such as when the background is also part of the signal), this can alert the model designers regarding the existence of spurious correlations in the dataset, and highlight the need to collect a better dataset.
> > > > >
> > > > > ---
> > > > >
> > > > > *“I was just wondering whether there is a realistic scenario where such a decomposition exists in practice, since most datasets are at least weakly correlated in all features (with spurious ones such as backgrounds being one of the most prominent examples from the literature).”*
> > > > >
> > > > > Good question! The core question here is whether real datasets always have spurious correlations across the entire input because the existence of such extreme spurious correlations implies a vacuous signal-distractor decomposition. While this is difficult to answer in general, we believe that given a large enough dataset, non-vacuous decompositions are likely to exist because a large portion of spurious correlations are eliminated when the dataset is large and diverse. Moreover, we are currently unaware of methods to experimentally recover these decompositions, beyond VerT.
> > > > >
> > > > > Another possibility is to relax the definition of the signal-distractor decomposition such that the distractor is “weakly” correlated to the label rather than being completely uncorrelated. In such a case, one can use the strength of correlation rather than its presence or absense to define signal and distractor. However it remains to be seen how theoretically feasible such a definition would be. If successful, such a relaxation can help define a non-vacuous decomposition that applies more widely.
> > > > >
> > > > > ---
> > > > >
> > > > > *“For example, I'm still not entirely sure that the chosen subset of CelebA has a nonvacuous decomposition.”*
> > > > >
> > > > > For our CelebA experiments, we agree in principle that there could very well be some unintended spurious correlations that we did not foresee and that our method did not recover, leading to a vacuous decomposition. However, our pixel perturbation results show that this is likely not the case. For example, in Figure 3 we were able to mask out about 90% of the total “unimportant” pixels in the image without the model performance degrading. We also perform experiments on masking the signal instead of the distractor, by masking out “important” pixels as opposed to the unimportant ones. With this experiment, we get:
> > > > >
> > > > > | Fraction Masked | Acc, Masked Distractor | Acc, Masked Signal | Acc, Random Mask |
> > > > > | --- | --- | --- | --- |
> > > > > | 0.2 | 0.971 | 0.772 | 0.965 |
> > > > > | 0.4 | 0.967 | 0.569 | 0.935 |
> > > > > | 0.6 | 0.967 | 0.392 | 0.826 |
> > > > > | 0.8 | 0.960 | 0.339 | 0.587 |
> > > > >
> > > > > These results indicate that for the VerT models, there do not exist pixels outside the shown signal regions in Figure 2 (for example) that contain information about the label. These results show that the signal-distractor decomposition for CelebA is very likely non-vacuous. We shall discuss these in our updated draft.
> > > > >
> > > > > ---
> > > > >
> > > > > *As for the optimal predictors: Thank you for explaining. This makes more sense to me---I previously thought the "this" in the last sentence referred to only the immediately preceeding sentence and not the entire procedure. Would probably be good to make it less ambiguous in the revision.*
> > > > >
> > > > > Glad that our explanation helped! We will definitely change the phrasing to make it less ambiguous.
> > > > >
> > > > > Thanks again for the responses! If you believe your concerns have been adequately addressed, we would ask you to consider raising your score.

---

### Author Rebuttal · Authors · 2023-08-09

We would like to thank all the reviewers for their constructive feedback. We are glad that reviewers found our theory “simple yet sound” (reviewer qGoL) and that our paper had “solid empirical evaluation” (reviewer dxrg). However, we feel that there were also some misunderstandings with respect to our method, and we aim to clarify these here.

**(G1) The core idea of the paper:** We’d like to perform feature attribution of black-box model f using a **masking-based procedure** (i.e., QFA in definition 1, with some input-independent choice of Q). However, this is problematic, as the **underlying black-box model may not be robust to perturbations** introduced by our procedure, leading to large change in model outputs upon masking. Ideally, one needs to find a distribution Q that the model is maximally robust to (if that even exists) and use that choice of Q for QFA to perform attribution. Another solution, which we propose, is to assume a choice of Q, and **fine-tune the black-box model to be robust to masking made with this Q** (also called the Q-verifiable model in definition 2) in the non-important features.

In particular, Q-verifiability can be thought of as a very specific form of robustness, where models are ONLY robust wrt the distractor in the input and NOT the signal, thus distinguishing it from classical robustness. Further, verifiability requires robustness wrt masking, as opposed to that additive noise, as is usual with the robustness literature. However, note that we do not know the signal and distractor apriori, which is why we use alternating minimization in VerT to alternately estimate the signal-distractor masks; and the resulting Q-verifiable models. We shall emphasize this point of view of verifiability as “robustness to masking on the distractors”, in the main paper.

**(G2) How is this paper different from prior works on feature attribution?** Classical feature attribution methods (LIME, SHAP, smoothgrad, etc) work by attributing to black-box models. As we mention above, these may lead to erroneous results when the model is non-robust to the perturbations introduced by these methods. On the other hand, our method (VerT + QFA) involves adapting the model to the attribution method, making this fundamentally distinct from classic feature attribution methods. Thus our framework contains aspects of both inherently interpretable models and post-hoc explanations.

Another fundamental contribution of this work lies in its conceptual framework. Classical feature attribution literature does not have a precise notion of “ground truth”, making it fundamentally unclear what quantities these are estimating in the first place. In this paper, **we introduce one notion of ground truth – the signal-distractor decomposition** that is a property of the dataset itself (and NOT the model!). Using this notion, we are able to verify that our procedures work as intended at least in settings where the ground truth is known, and when the model correctly identifies the entire signal component (i.e., for well-performing models).


**(G3) Why is theorem 1 interesting?** Theorem 1 states that QFA (with some choice of Q) used with an appropriate Q-verifiable model (using the same Q distribution) is able to recover the underlying signal-distractor decompositions for optimal models on a dataset. This is interesting because this works with any choice of Q! This should intuitively make sense: if a model is able to be robust to masking with some distribution Q, then we can use this fact to do feature attribution, for any chosen Q.

**(G4) Does VerT only work in limited settings?** Not at all! Theorem 1 shows that when we are able to define the ground truth, i.e., the signal-distractor decomposition, this is recovered by VerT. This does not imply that VerT does not apply in other settings. In other settings where the signal-distractor distribution is unknown we only cannot evaluate the correctness of the QFA mask w.r.t. ground truth, but the theorem 1 guarantees correctness for optimal models. Non-optimal models may not identify the correct signal component (hence their non-optimality), making it such that the signal cannot be theoretically extracted from such models. However, in practice if a model is close-to-optimal, then it is reasonable to assume that it identifies close-to-correct signal components, and this is exactly what our experiments demonstrate. VerT models recover masks that are close to the ground truth, and our method (QFA) is able to identify this.

We aim to add discussions in the paper clarifying points (G1-G4).

**(G5) Additional Experiments:** Please refer to attached pdf.
As requested by reviewers QRQ5, 2ozo, and qGoL, we add SHAP as a baseline. We note that in general SHAP does not often perform as well as most gradient-based methods for image data. We find that it performs similarly to random attributions.

We also add B-CosNets as an inherently-interpretable model baseline, as suggested by reviewer qGoL. We find that B-CosNets performs comparably to our method for the IOU test on Hard MNIST. We were unable to train it to convergence on the Chest X-ray dataset. We also find that visualizations created by B-CosNets align with our expectations and appear visually interpretable. However, our method still significantly outperforms B-CosNets on the pixel perturbation test, showing that B-CosNets are not robust to perturbations of the distractor (i.e. are not verifiable). We also note that our method can be applied to a trained black-box model of any architecture and training procedure, whereas B-CosNets, like all inherently interpretable models, cannot.

We finally perform the ablation requested by reviewer QRQ5 to test the effect that the choice of Q has empirically. We consider various parameterizations of the normal distribution used for Q, and find that results are relatively consistent across the different choices of Q.

---

### Decision · Program_Chairs · 2023-09-21

**Decision:**

Accept (poster)

**Comment:**

The authors propose a framework for verifiable feature attribution, which, according to the authors, fixes some issues with previous definitions of feature attribution. In particular, the proposed definition depends on a counterfactual distribution that is used to replace features of interest, and quantify how the output changes. The paper demonstrates how verifiable feature attribution cannot be achieved in black box models. The authors propose a new method that transforms a black box model into one where feature attribution can be verified. The authors run experiments demonstrating the success of this method on synthetic and real world tasks.

The authors study an important problem. This publication may spur further research in the direction of converting a black box model to an interpretable one. Drawbacks identified by the reviewers include: some assumptions about the data seem unrealistic; the method is developed to optimize a particular criteria and then it is compared against other methods that do not optimize for this criteria, thus resulting in a somewhat circular comparison scheme; as a result, the evaluation does not seem fair or appropriate; the algorithm cannot yet scale to larger models and datasets; there is some disagreement between what the authors claim what VerT does (the model, after "conversion" is equivalent to the original model in function space) and some empirical results showing that VerT produces models that are more robust. I would suggest discussing some of these drawbacks among limitations, and, when possible, address them directly when updating the paper.